# A beneficial adaptive role for CHOP in driving cell fate selection during ER stress

Kaihua Liu[1], Chaoxian Zhao [2], Reed C Adajar[3], Diane DeZwaan-McCabe[3] & D Thomas Rutkowski [1,3,4✉]

## Abstract

**Cellular stresses elicit signaling cascades that are capable of either mitigating the inciting dysfunction or initiating cell death. During endoplasmic reticulum (ER) stress, the transcription factor CHOP is widely recognized to promote cell death. However, it is not clear whether CHOP also has a beneficial role during adaptation. Here, we combine a new, versatile, genetically modified *Chop* allele with single cell analysis and with stresses of physiological intensity, to rigorously examine the contribution of CHOP to cell fate. Paradoxically, we find that CHOP promotes death in some cells, but proliferation—and hence recovery—in others. Strikingly, this function of CHOP confers to cells a stress-specific competitive growth advantage. The dynamics of CHOP expression and UPR activation at the single cell level suggest that CHOP maximizes UPR activation, which in turn favors stress resolution, subsequent UPR deactivation, and proliferation. Taken together, these findings suggest that CHOP's function can be better described as a "stress test" that drives cells into either of two mutually exclusive fates—adaptation or death—during stresses of physiological intensity.**

**Keywords** Adaptation; Cell Fate; ER Stress; Cell Proliferation; Unfolded Protein Response
**Subject Categories** Autophagy & Cell Death; Molecular Biology of Disease; Post-translational Modifications & Proteolysis

## Introduction

The unfolded protein response (UPR) promotes the restoration of homeostasis during ER stress, when the ER is burdened by an excess of client proteins beyond the organelle's capacity to fold them. Deletion of individual signaling components of the UPR compromises embryonic development and hastens the progression of numerous mouse models of diseases including neurodegenerative, metabolic, cardiovascular, and neoplasmic disorders, suggesting that the response is, on balance, beneficial to health. From these

findings, it can be assumed that, even though UPR activation and, often, ER structural disruption are associated with many human diseases as well (Lindholm et al, 2017), these pathologies would be even worse were it not for the protective action of the UPR.

Despite its beneficial impacts on cellular physiology, the UPR, like other stress responses, can initiate cell death pathways when homeostasis cannot be restored. While numerous pathways connecting UPR activation to cell death have been described (Hetz & Papa, 2018), the first identified in mammals, and most characterized, is incited by CHOP (C/EBP-homologous protein). Cells lacking CHOP undergo far less cell death in vitro when challenged with ER stress-inducing agents than do wild-type cells (Hu et al, 2018; Zinszner et al, 1998). Likewise, whole-body deletion of CHOP protects against many—though, notably, not all—experimental pathologies with an ER stress component including diabetes, cancers, liver injury, and neurodegeneration, while simultaneously diminishing the expression of cell death markers (Yang et al, 2017). These findings have led to the widely accepted idea that the primary (or perhaps only) cellular consequence of CHOP activity is the promotion of cell death during excessive ER stress, and that this cell death accelerates the progression of diseases exacerbated by ER stress.

Yet the molecular actions of CHOP are at least superficially at odds with its role in promoting cell death. CHOP is not a death effector in the traditional sense; it is not a BCL2 family member nor does it interact directly with caspase cascades. Rather, it is a transcription factor of the C/EBP family, the members of which are involved in the regulation of proliferation, differentiation, immune responses, and metabolism (Johnson, 2005; Nerlov, 2007). Indeed, the first molecular function ascribed to CHOP was as a dominant-negative regulator of C/EBPα and C/EBPβ (Ron and Habener, 1992). Since that time, it has been proposed that CHOP regulates expression of BCL2 family members including BCL2 itself (McCullough et al, 2001) and BIM (Puthalakath et al, 2007) as well as the DR5 member of the pro-apoptotic Tumor Necrosis Factor Receptor superfamily (Yamaguchi and Wang, 2004). CHOP also regulates ERO1α, an ER oxidoreductase that has been postulated to hasten cell death through overexuberant disulfide bond formation in the ER lumen, which is accompanied by calcium dysregulation and increased production of reactive oxygen species (Li et al, 2009; Song et al, 2008). However, the most exhaustive studies to date have shown that, at least in mouse embryonic fibroblasts (MEFs)—the cell type of choice for basic cellular studies of ER stress signaling—CHOP enhances

[1]Interdisciplinary Graduate Program in Human Toxicology, University of Iowa Carver College of Medicine, Iowa City, IA, USA. [2]Shanghai Cancer Institute, Renji Hospital Affiliated to Shanghai Jiao Tong University School of Medicine, Shanghai, China. [3]Department of Anatomy and Cell Biology, University of Iowa Carver College of Medicine, Iowa City, IA, USA. [4]Internal Medicine, University of Iowa Carver College of Medicine, Iowa City, IA, USA. ✉E-mail: thomas-rutkowski@uiowa.edu

protein synthesis through its upregulation of tRNA synthetase genes and of GADD34 (Growth Arrest and DNA Damage) (Han et al, 2013; Krokowski et al, 2013; Marciniak et al, 2004). The latter dephosphorylates the translation initiation factor eIF2α, promoting the restoration of protein synthesis that had been inhibited by eIF2α phosphorylation during ER stress by the sensor PERK (PKR-like ER kinase) (Novoa et al, 2001) or by other eIF2α kinases during other stresses. eIF2α phosphorylation inhibits general translation but stimulates translation of the transcription factor ATF4 (Harding et al, 2000) and of CHOP (Palam Baird and Wek, 2011), while ATF4 transcriptionally upregulates CHOP (Ma et al, 2002). CHOP and ATF4 in turn cooperate to induce GADD34 (Han et al, 2013; Marciniak et al, 2004), which is itself translationally stimulated by eIF2α phosphorylation (Lee Cevallos and Jan, 2009), completing a negative feedback loop (Brush Weiser and Shenolikar, 2003; Kojima et al, 2003; Novoa et al, 2001). Thus, CHOP would seem to have a fundamental role governing the temporal dynamics of the stress response.

Similarly, the dynamics of CHOP expression are difficult to reconcile with a strictly death-promoting function. It is unquestionably true that CHOP is robustly upregulated by experimental ER stresses and promotes cell death in those contexts. However, most such stresses are sufficiently severe that cell death is effectively an obligate outcome, and they are probably poor proxies for the stresses of more physiological intensity that the UPR evolved to protect against. Even in chronic disease states such as diabetes and neurodegeneration where ER stress-induced cell death has been implicated, very few cells are dying at any given time. The UPR can also be induced experimentally by stresses sufficiently mild that few if any cells die, and under conditions when cells like MEFs actively proliferate despite ongoing or recurrent ER stress (Rutkowski et al, 2006). Therefore, CHOP induction is not fundamentally incompatible with cellular adaptation and proliferation, and it might in fact have a beneficial role in promoting adaptation that has heretofore gone unappreciated. Yet arguing against this possibility is the observation that animals lacking CHOP have no apparent basal phenotype, and cells are not obviously impaired by its loss—even restoration of protein synthesis during stress occurs without CHOP due to the actions of the constitutive eIF2α phosphatase CReP (Constitutive Repressor of eIF2α Phosphorylation) (Jousse et al, 2003). Thus, whether CHOP has functional roles beyond promoting cell death is unclear.

In this paper, we describe the creation of a new, versatile, genetically modified allele of *Chop* that has led us to rigorously uncover a dual role for CHOP in driving cells into either of two mutually exclusive fates—death or proliferation—during ER stress.

## Results

### CHOP supports EdU incorporation during ER stress in primary mouse embryonic fibroblasts

C/EBP-family transcription factors generally stimulate differentiation and inhibit proliferation (Johnson, 2005; Nerlov, 2007), and CHOP likely acts as a dominant-negative member of this family (Ron and Habener, 1992). Thus, despite CHOP's characterization as growth arrest-inducible and its putative role in cell cycle arrest (Harris et al, 2001; Hendricks-Taylor and Darlington, 1995; Mihailidou et al, 2010), we speculated that CHOP might have an unappreciated role in proliferation that might ordinarily be

obscured by the acutely cytotoxic doses of chemical agents typically used to elicit ER stress and probe CHOP's function. To test this idea, we compared the incorporation of the thymidine analog 5-ethynyl-2'-deoxyuridine (EdU) in synchronized wild-type (w.t.) and *Chop* knockout (*Chop*$^{-/-}$) primary mouse embryonic fibroblasts (MEFs) that were treated with either 2.5 or 50 nM thapsigargin (TG), a stressor that causes ER stress by blocking ER calcium reuptake. The doses used were one- to two orders of magnitude lower than typically used to elicit ER stress, yet are still sufficient to activate the UPR. Moreover, as we have shown, MEF cultures can still undergo net proliferation during treatment with 2.5–10 nM TG despite the perturbation (Rutkowski et al, 2006).

The cells used for this experiment were isolated by timed intercross of animals heterozygous for a previously described constitutive *Chop* null allele (Zinszner et al, 1998). To ensure that any results were robust and attributable to the presence or absence of CHOP and not to unrelated line-to-line differences, experiments were performed in at least two separate wild-type (w.t.) and *Chop*$^{-/-}$ lines. Under these conditions, there was no apparent difference in EdU incorporation between w.t. and *Chop*$^{-/-}$ cells in the absence of stress, indicating that knocking out CHOP does not affect basal proliferation (Figs. 1A and EV1A). In this experiment, treatment with 2.5 nM TG trended toward slightly diminished EdU incorporation in *Chop*$^{-/-}$ cells but not in wild-type cells—though this trend did not reach statistical significance—while 50 nM TG almost entirely prevented EdU incorporation in *Chop*$^{-/-}$ cells yet allowed for modest but significant incorporation in w.t. cells (Fig. 1B). Comparable results were observed in both synchronized (Fig. 1B) and non-synchronized cells (Fig. EV1B). A similar phenotype was observed when MEFs were treated with the mechanistically unrelated ER stressor tunicamycin (TM, Figs. 1C and EV1C), suggesting that the CHOP-mediated EdU incorporation difference is not stressor-specific. Because EdU incorporation reads out on S-phase progression, these data suggest that CHOP promotes cell proliferation. We found that deletion of CHOP increased the percentage of cells in G2 at the expense of G1 cells (Fig. EV1D). There was also decreased mRNA expression of the cell cycle regulators *Pcna*, *Cyclin A2*, and *Cyclin E1*, and increased expression of the negative regulator of PCNA *p21*, although *Ki67* and *Cdk6* expression were also elevated (Fig. EV1E). These data are consistent with CHOP facilitating cell proliferation but not acting as a direct regulator of cell cycle progression. The difference in proliferation was also observed in the liver cell line TIB-73 when CHOP was deleted using CRISPR/Cas9 targeting (Figs. 1D and EV1F); therefore, the phenotype is not cell-type specific. The effect was also observed in MEFs after immortalization by large T-antigen (Fig. EV1H), but not in the immortal MEF line 3T3 (Fig. EV1I), indicating that there are at least some scenarios where the difference is lost. A potential role for CHOP in promoting proliferation was first tentatively observed but otherwise unexplored in the original characterization of *Chop*$^{-/-}$ cells and animals (Zinszner et al, 1998).

### CHOP is necessary and sufficient to support EdU incorporation during ER stress

The difference in EdU incorporation between w.t. and *Chop*$^{-/-}$ cells might reflect a direct role for CHOP in proliferation, or a compensatory mechanism that arises in embryos in which CHOP has been deleted. Thus, we created an allele in which CHOP could be controlled with temporal precision. In this allele, termed "*FLuL*" (*Frt-Luciferase-LoxP*), a gene trap cassette expressing a splice acceptor, nanoLuciferase (nLuc), and a transcriptional terminator, together flanked by *Flp* sites, blocks expression of the CHOP open

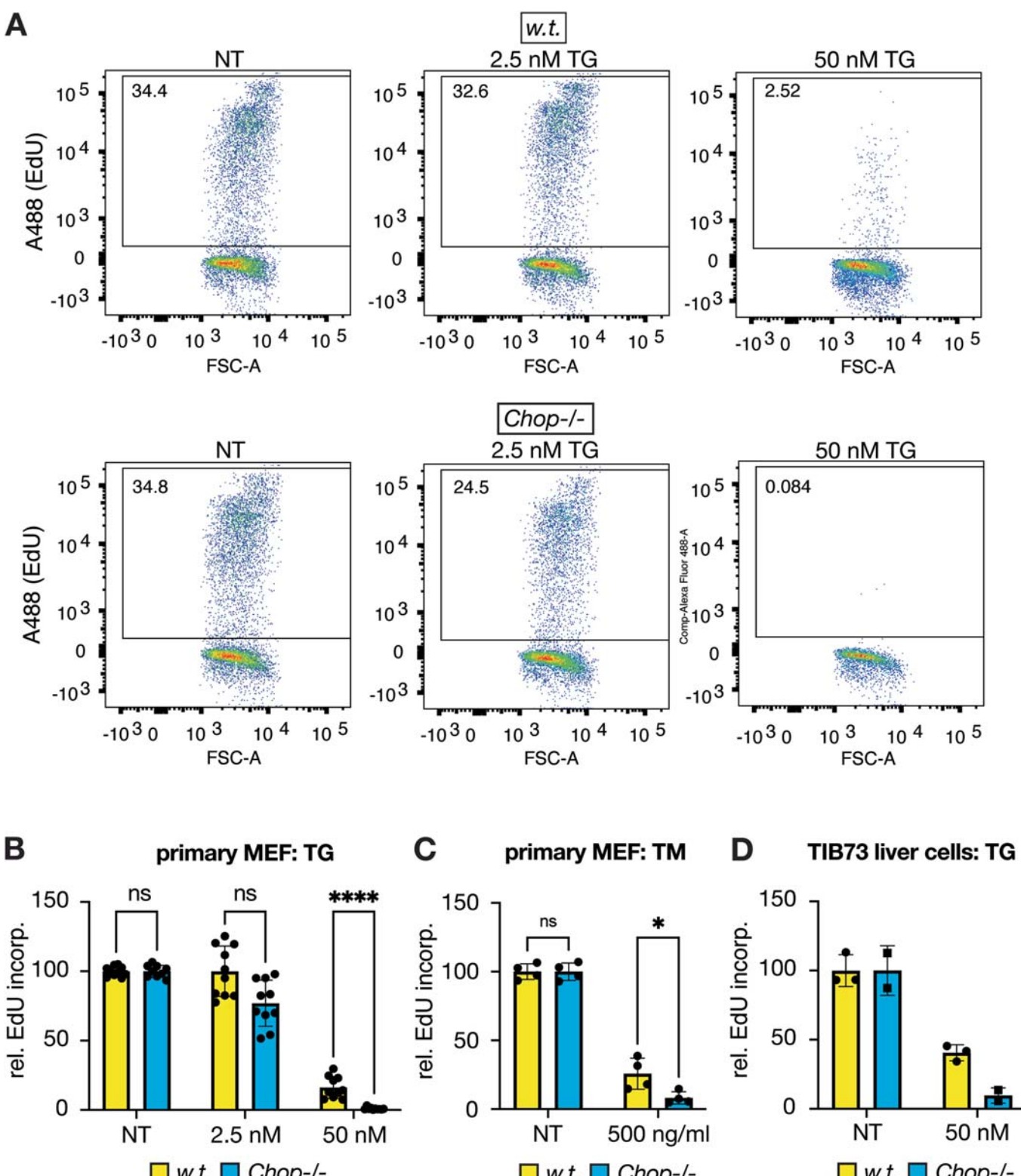

reading frame, which resides in *Chop* exons 3 and 4 (Fig. 2A). Deletion of the cassette by the FLPo recombinase creates a floxed allele ("*fl*") that restores CHOP expression (Fig. 2B), which can then be again eliminated by the action of CRE ("*KO*") (Fig. 2C). Consistent with previous findings with global CHOP knockout

mice (Zinszner et al, 1998), *Chop^FLuL/FLuL*^ mice were viable, fertile and grossly indistinguishable from wild-type (w.t.) mice. Primary *Chop^FLuL/FLuL*^ MEFs expressed nLuc, and its expression was greatly enhanced by TM (Fig. 2D), which was expected since nLuc should be under the same transcriptional and translational control as

Figure 1. CHOP supports EdU incorporation during ER stress in cells.

(A) Wild-type and *Chop−/−* mouse primary embryonic fibroblasts (MEFs) were serum starved and then treated in complete media with vehicle (NT) or thapsigargin (TG, 2.5 nM or 50 nM) for 24 h. EdU was added directly to culture media 6 h before cell harvest, and EdU incorporation was analyzed by flow cytometry. Representative plots are shown. (B) Quantification of relative EdU incorporation in primary MEFs after TG treatment from (A). Data were normalized relative to the EdU positive percentage in untreated cells of each genotype. Data were aggregated from two experiments, using two separate w.t. and *Chop−/−* lines. Statistical analysis was by Mann–Whitney because the data did not satisfy the normality criterion for ANOVA. Data are means +/− S.D.M. from independently treated wells. ns = not statistically significant; ****$p < 0.0001$. (C) Similar to (B) except treatment with 500 ng/ml TM or vehicle (NT). Means +/− S.D.M. from independently treated wells. Two-way ANOVA with Šidák, ns = not significant, *$p < 0.05$. (D) Relative EdU incorporation in control wild-type or CRISPR-targeted *Chop−/−* TIB-73 liver cells after TG. Each data point represents results from an independent clone (3 wild-type and 2 knockout). Source data are available online for this figure.

CHOP itself. At the same time, CHOP expression was completely lost in *Chop^FLuL/FLuL* MEFs but restored in *Chop^fl/fl* MEFs in which the gene trap had been deleted by FLPo (Fig. 2E). The *FLuL* allele could be manipulated both in vivo and in vitro—in the latter case by adding recombinant adenovirus expressing FLPo to restore CHOP in the *FLuL* allele, or expressing CRE to delete CHOP in the floxed allele. In either case, Ad-GFP is used as a control. Demonstrating the feasibility of this approach, treatment of *Chop^FLuL/FLuL* cells with Ad-FLPo eliminated transcripts expressing nLuc and restored transcripts expressing *Chop* exons 3–4; as expected, all transcripts from the *Chop* allele were stress-dependent in their expression (Fig. 2F). Stress-mediated induction of *Chop* exons 1–2 (which are represented in both the *Chop* transcript and the *nLuc* transcript) was actually higher in cells treated with FLPo (Fig. 2F, top), which implies either that the *nLuc* transcript is degraded more rapidly than is the *Chop* transcript, or that CHOP contributes to its own regulation such that restoration of CHOP protein expression by FLPo enhances transcription from the locus (or both). This in vitro deletion approach has the advantage of allowing the effects of CHOP to be examined in cells of the same origin, which is not the case when using MEF lines derived from separate embryos isolated from heterozygote intercrosses.

Further validating these lines, w.t. and *Chop^FLuL/FLuL* MEFs were treated with different concentrations of TG for either 24 or 36 h. PARP cleavage, an indicator of apoptotic cell death, was more prominent in w.t. MEFs than in *Chop^FLuL/FLuL* MEFs, particularly at 50 nM TG (Figs. 2G and EV1G). Similarly, FLPo-infected (and hence wild-type) MEFs from the *Chop^FLuL/FLuL* background released more lactate dehydrogenase (LDH) than GFP-infected MEFs after TG treatment, suggesting that FLPo-directed CHOP re-expression in *Chop^FLuL/FLuL* MEFs enhanced cell death under ER stress (Fig. 2H). Taken together, these data validate in this new model the well-described role of CHOP in promoting cell death upon ER stress.

The CHOP-dependent effect on EdU incorporation observed in Fig. 1 was also seen in MEFs harboring this new *Chop* allele. Wild-type or *Chop^FLuL/FLuL* MEFs were treated with 50 nM TG or 500 ng/mL TM for 24 h, and EdU incorporation was evaluated by flow cytometry. As in Fig. 1, no consistent difference in basal EdU incorporation was seen between w.t. and *Chop^FLuL/FLuL* MEFs. However, there was more EdU incorporation in w.t. MEFs compared to *Chop^FLuL/FLuL* MEFs during both TG and TM treatment (Fig. 3A,B). Induction of CHOP expression in *Chop^FLuL/FLuL* MEFs by provision of Ad-FLPo was sufficient to increase EdU incorporation upon TG treatment (Figs. 3C and EV2A), while deletion of CHOP by provision of Ad-CRE to *Chop^fl/fl* MEFs decreased EdU incorporation (Figs. 3D and EV2B). Taken together,

these results robustly establish that CHOP is both necessary and sufficient to support EdU incorporation under ER stress.

## CHOP supports proliferation during ER stress through restoration of protein synthesis

Given the well-established role of CHOP in promoting cell death, we examined the relationships among EdU incorporation, cell cycle progression, and cell death. First, double staining for EdU and the cell death marker cleaved PARP showed the EdU-positive and dying populations of cells to be essentially mutually exclusive (Fig. 4A). In addition, the pan-caspase inhibitor Q-VD-OPH did not alter the propensity of wild-type cells to incorporate EdU over *Chop^FLuL/FLuL* MEFs (Fig. 4B). Therefore, EdU-positive cells do not represent a population of dying cells, nor are they affected by the ability of other cells to die. Rather, EdU-positive cells progress through the cell cycle. To demonstrate this, we utilized the reversible ER stressor-dithiothreitol (DTT). Cells were pulsed with 5 mM DTT for 2 h, EdU was incorporated after DTT washout by pulse labeling, and the cell cycle progression of the EdU-positive cells over time was assessed by propidium iodide (PI) staining of DNA content. As we have shown, cells can survive and proliferate despite repeated pulses with high concentrations of DTT (Wu et al, 2007). EdU-positive cells progressed from S-phase through G2/M such that, by 12 h after the EdU pulse, almost all of the EdU-positive cells had returned to the 2 N DNA content characteristic of G1 (Figs. 4C and EV3A–D). This was true of both w.t. and *Chop^FLuL/FLuL* EdU-positive cells, though as expected there were far fewer of the latter. We also considered the possibility that dying cells release some mitogenic factor that would stimulate proliferation of non-dying cells independent of the receiving cells' *Chop* genotype (Willy et al, 2015). To test this idea, we carried out a medium switch experiment where the culture medium of TG-treated w.t. or *Chop^FLuL/FLuL* MEFs was collected and applied to non-treated w.t. MEFs, in which EdU incorporation was then assessed. Under these conditions, whether the media came from w.t. or *Chop^FLuL/FLuL* MEFs had no effect on EdU incorporation in the receiving cells (Fig. EV3E). Taken together, these data show that CHOP promotes cell cycle progression and argue that this effect does not likely arise as an indirect consequence of CHOP-mediated cell death.

CHOP has been proposed to transcriptionally regulate apoptosis (Han et al, 2013; Hu et al, 2018; Krokowski et al, 2013; Zinszner et al, 1998), inflammation (DeZwaan-McCabe et al, 2013; Scaiewicz et al, 2013; Willy et al, 2015), and lipid metabolism (Chikka et al, 2013) among other processes. While the functions of CHOP might depend on the cell type in which it is expressed and the complement of other transcription factors with which it might interact, in MEFs at least, its essential impact on gene expression

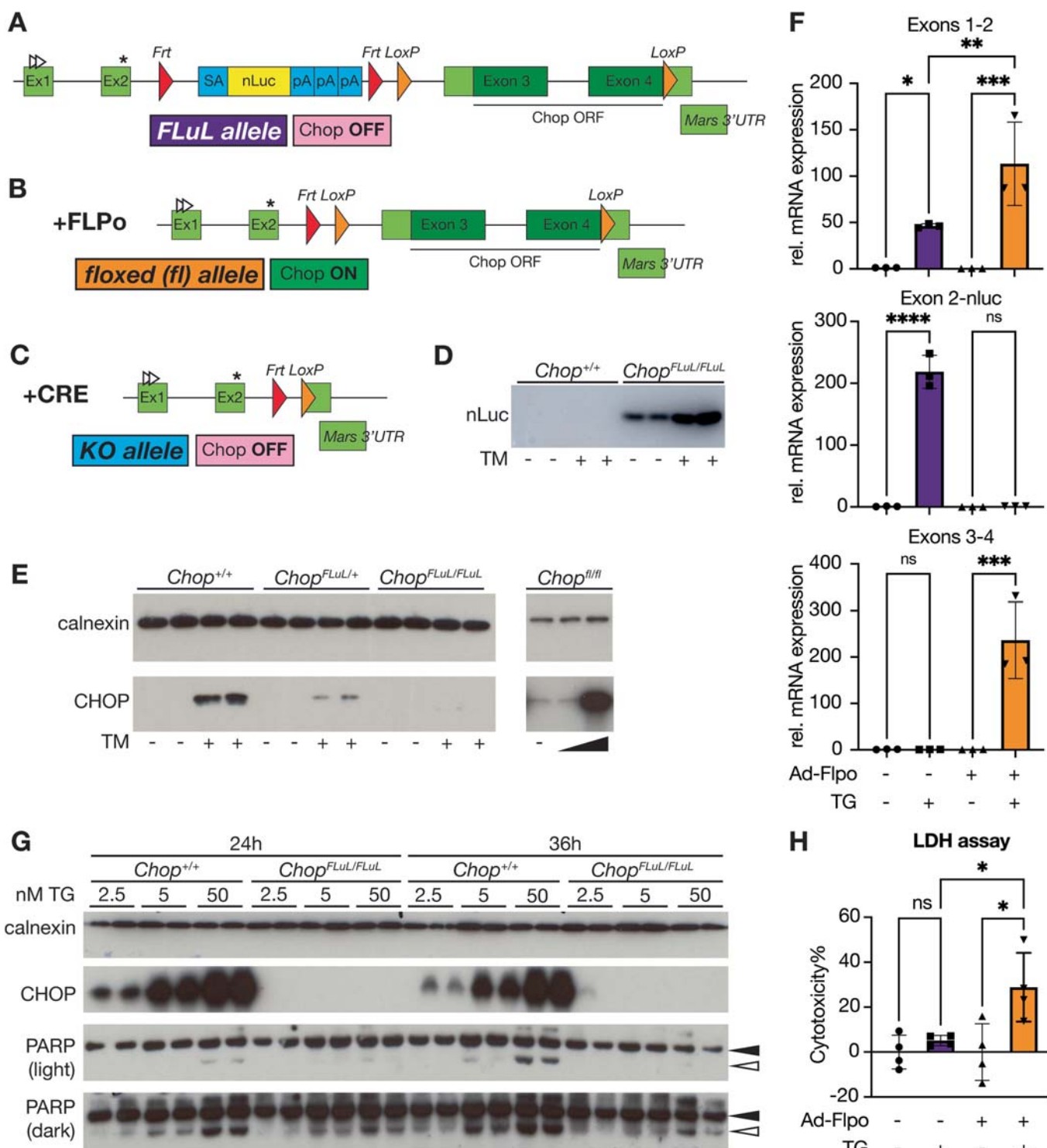

appears to be the restoration of protein synthesis, which occurs both through CHOP-dependent upregulation of the eIF2α phosphatase GADD34 (Marciniak et al, 2004) and the broader induction of tRNA synthetases and other genes that augment the protein biosynthetic capacity (Han et al, 2013; Krokowski et al, 2013). CHOP-dependent enhancement of protein synthesis was shown to contribute to cell death under stress by enhancing the

burden of nascent proteins on an already-stressed ER (Marciniak et al, 2004). However, this effect was tested under stress conditions severe enough that cells would have had little likelihood of ever restoring ER homeostasis. Thus, we considered that augmenting protein synthesis might benefit cells that are able to restore ER homeostasis with a proliferative advantage. To test this idea, we took advantage of ISRIB (Integrated Stress Response Inhibitor), a

Figure 2. **Creation and validation of a new targeted *Chop* allele.**

(A–C) Graphic illustrations of *Chop* FLuL (A), *fl* (B), and KO (C) alleles, with the expression status of CHOP shown underneath each. Throughout Figs. 2–5, yellow (wild-type), purple (*FLuL*), orange (*fl*), and cyan (KO) indicate which alleles are being examined in bar graphs. The CHOP ORF, localized entirely within exons 3 and 4, is indicated. The white arrowheads and asterisk represent start and stop codons, respectively, for the *Chop* upstream open reading frame (uORF). (D) In-gel nanoLuciferase detection after SDS-PAGE in protein lysates from w.t. and *Chop*^FLuL/FLuL^ primary MEFs after 5 µg/mL TM treatment for 4 h. (E) Immunoblot showing CHOP expression in w.t., *Chop*^FLuL/+^, and *Chop*^FLuL/FLuL^ primary MEFs treated with vehicle (NT) or 5 µg/mL TM for 4 h (left), or protein lysates from *Chop*^fl/fl^ MEFs treated with vehicle (NT), 100 ng/mL, or 5 µg/mL TM for 24 h. (F) CHOP-null or wild-type MEFs were generated by infecting *Chop*^FLuL/FLuL^ MEFs with Ad-GFP or Ad-GFP-FLPo, sorting out GFP-positive cells, expanding different cell poupulations, and confirming CHOP re-expression or lack thereof. RNAs were isolated after 500 nM TG treatment for 4 h. Expression of different splicing variants was measured by qRT-PCR, with the recognized exons indicated. Means +/− S.D.M. from independently treated wells. One-way ANOVA with Tukey, ns = not significant, *$p < 0.05$; **$p < 0.01$; ***$p < 0.001$; ****$p < 0.0001$. (G) w.t. and *Chop*^FLuL/FLuL^ primary MEFs were treated with different doses of TG for 24 h or 36 h and western blot was used to assess the expression of CHOP and PARP (uncleaved and cleaved indicated by black and white arrowheads respectively). The expression of the ER resident protein calnexin, which is not affected by ER stress, was used as a loading control. (H) An LDH cytotoxicity assay was performed on media from Ad-GFP- or Ad-FLPo-infected *Chop*^FLuL/FLuL^ primary MEFs after vehicle or 500 nM TG treatment for 24 h. Means +/− S.D.M. from independently treated wells. One-way ANOVA with Tukey, ns = not significant, *$p < 0.05$. Source data are available online for this figure.

chemical that counteracts the suppression of protein synthesis under stress by allosterically modulating the GTP exchange factor eIF2B in such a way as to prevent its engagement by eIF2α (Sekine et al, 2015; Sidrauski et al, 2013; Zyryanova et al, 2021). Under these conditions, ISRIB dampened eIF2α-dependent signaling including expression of at least ATF4 and possibly CHOP as well (Fig. 4D). And, as expected, ISRIB largely or completely prevented a short treatment (4 h) of TG from inhibiting protein synthesis, as assessed by ³⁵S incorporation, and this effect was seen in both CHOP-expressing cells (*Chop*^fl/fl^ treated with Ad-GFP) and CHOP-null cells (*Chop*^fl/fl^ cells treated with Ad-CRE) (Fig. EV3F, G). ISRIB increased EdU incorporation upon TG treatment in CHOP-null cells, whereas it had no such effect in wild-type cells (Fig. 4E). Conversely, treatment of cells with the GADD34 inhibitor salubrinal, which perpetuates the stress-dependent inhibition of protein synthesis (Boyce et al, 2005), diminished EdU incorporation (Fig. 4E), as did siRNA-dependent knockdown of *Gadd34* (Fig. 4F,G). From these data, it is likely that the proliferative function of CHOP is carried out at least in part by its previously described role in reversing the attenuation of protein synthesis caused by eIF2α phosphorylation.

## CHOP confers a competitive advantage on cells under mild ER stress

The observation that CHOP promotes proliferation raises the possibility that, during stresses of mild intensity (which presumably better reflect the sorts of stresses encountered in normal physiological scenarios), CHOP might confer a functional benefit that has not been appreciated before. To test this prediction in the most sensitive and rigorous way, we subjected cells of identical origin either expressing or lacking CHOP to a growth competition assay. The experiment was performed in both directions: either *Chop*^FLuL/FLuL^ cells were treated with Ad-GFP or Ad-FLPo; or *Chop*^fl/fl^ cells were treated with Ad-GFP or Ad-CRE. We used quantitative PCR to identify each allele (Fig. 5A) and confirmed that each primer pair was both specific (Fig. 5B) and efficient (Fig. 5C) in detecting the allele against which it was designed. We then mixed each pair of cells in independent replicate plates at a 1:1 ratio and cultured them for up to 3 passages, in the presence of either vehicle or 2.5 nM TG, with the media and stressor refreshed every 48 h. The cells were passaged upon reaching confluence, with the replicates being kept separate from each other, and with an aliquot kept from each plate for qPCR. As we have previously shown,

2.5 nM TG does not appreciably diminish the proliferative capacity of MEFs, even though that dose is sufficient to activate the UPR (Rutkowski et al, 2006). When cells were cultured in stressor-free media, there was no significant change in the ratio of either the *FLuL*-to-*Flox* alleles or the *Flox* to KO alleles (Fig. 5D,E)— unsurprising since no detectable CHOP is expressed in the absence of stress (e.g., Fig. 4E). However, CHOP-expressing cells of both origins significantly outcompeted CHOP-null cells when challenged with 2.5 nM TG (Fig. 5D,E). These data suggest that CHOP augments the functional recovery of cells from or adaptation to mild ER stress.

## Single cell analysis confirms a proliferative role for CHOP

The fact that CHOP promotes both cell death and proliferation, in mutually exclusive populations, suggests that the response of a population of cells to a stressor is not uniform. Thus, we wanted to better understand how CHOP expression corresponded to cell fate in individual cells. To accomplish this, we used flow cytometry to detect endogenous CHOP expression in cells subjected to ER stress. For this purpose, we used a monoclonal antibody whose specificity for CHOP we have previously demonstrated (DeZwaan-McCabe et al, 2013). We treated cells with a dose of TG (5 nM) that we have previously shown permits net proliferation in MEFs (Rutkowski et al, 2006). Over a time course of that mild TG treatment, the expression of CHOP increased uniformly in the cell population, reaching its maximum by 8 h, at which point all cells expressed CHOP. However, at subsequent time points, the cells began to separate into two populations, one of which retained maximal CHOP expression and the other of which became CHOP-negative, with relatively few cells in an intermediate state (Fig. 6A). In contrast, cells treated with 100 nM TG rapidly became CHOP-positive, to the same extent as cells treated with 5 nM TG, but remained so through the time course (Fig. 6B). This behavior was mirrored in vivo, in animals challenged by an IP injection of TM. Eight hours after challenge, most if not all hepatocytes expressed CHOP, as seen by immunostained nuclei, whereas at a later time point a few cells remained strongly CHOP-positive while CHOP was undetectable in the remainder (Fig. 6C). Under conditions of severe stress (100 nM), there was little or no effect of CHOP on the overall ER stress burden, as assessed by splicing of the IRE1α target *Xbp1*; however, during mild stress (5 nM), CHOP enhanced UPR activation at the same time points (particularly 16 h) at which the population of cells had split (Fig. 6D, E). Thus, CHOP can promote

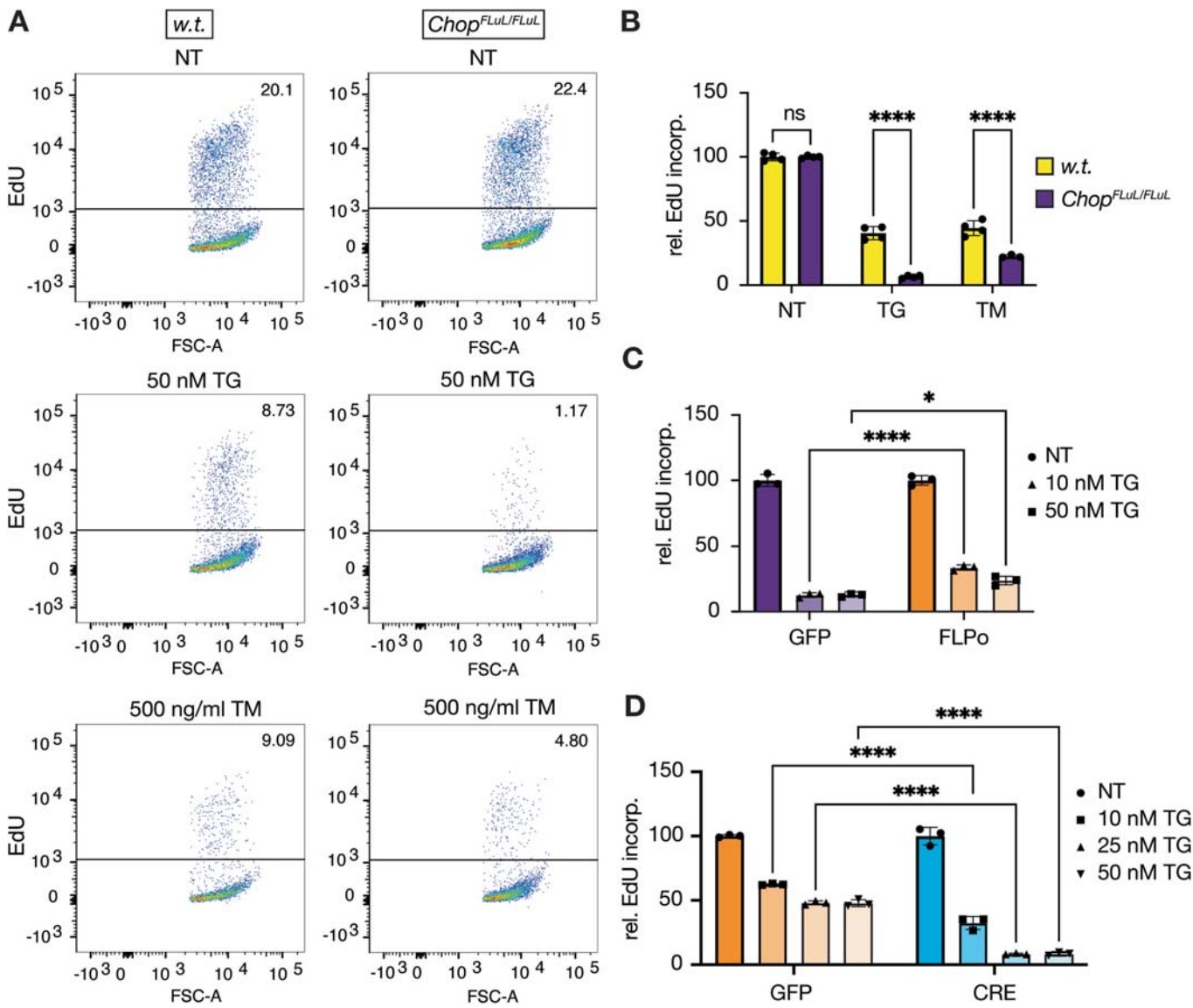

**Figure 3. CHOP is necessary and sufficient to support EdU incorporation during ER stress.**

(A) Representative flow cytometry images showing EdU incorporation in w.t. and *Chop^{FLuL/FLuL}* primary MEFs treated with vehicle (NT) or ER stressors (50 nM TG or 500 ng/ml TM) for 24 h. EdU was added 4 h before cell harvest. (B) Quantification of EdU incorporation of data from the experiment shown in (A) as in Fig. 1. Means +/− S.D.M. from independently treated wells. Two-way ANOVA with Šidák, ns = not significant; ****$p < 0.0001$. (C) EdU incorporation in Ad-GFP- or Ad-GFP-FLPo-infected *Chop^{FLuL/FLuL}* primary MEFs after 24 h of TG treatment. Means +/− S.D.M. from independently treated wells. Two-way ANOVA with Šidák, ns = not significant; *$p < 0.05$; ****$p < 0.0001$. (D) EdU incorporation in Ad-GFP- or Ad-GFP-CRE-infected *Chop^{fl/fl}* primary MEFs after 24 h of TG treatment. Means +/− S.D.M. from independently treated wells. Two-way ANOVA with Šidák, ****$p < 0.0001$. Source data are available online for this figure.

at least two mutually exclusive cell fates (death and proliferation) and is expressed in at least two distinct temporal patterns (transient and persistent).

These findings raise the question of whether the transience of CHOP expression has any bearing on whether or not cells proliferate. Unfortunately, because CHOP is a nuclear antigen, flow cytometry requires fixation and permeabilization, thus preventing meaningful downstream analysis of how these two populations of cells differ. Moreover, CHOP expression is extensively regulated transcriptionally, post-transcriptionally, and translationally, and probably by its degradation (Ma et al, 2002;

Palam Baird and Wek, 2011; Rutkowski et al, 2006; Ubeda et al, 1999; Zinszner et al, 1998), meaning exogenous fluorescent reporters are unlikely to faithfully reflect the dynamics of endogenous CHOP. Indeed, even our nLuc gene trap, though it is knocked into the *Chop* locus and (presumably) subject to the same transcriptional and translational control as is endogenous CHOP, shows a considerable basal expression that CHOP itself does not (Fig. 2D,E).

Thus, as an alternate approach, we used single-cell RNA-seq (scRNA-seq) to better characterize how *Chop* expression relates to cell fate, while recognizing that mRNA expression is an imperfect

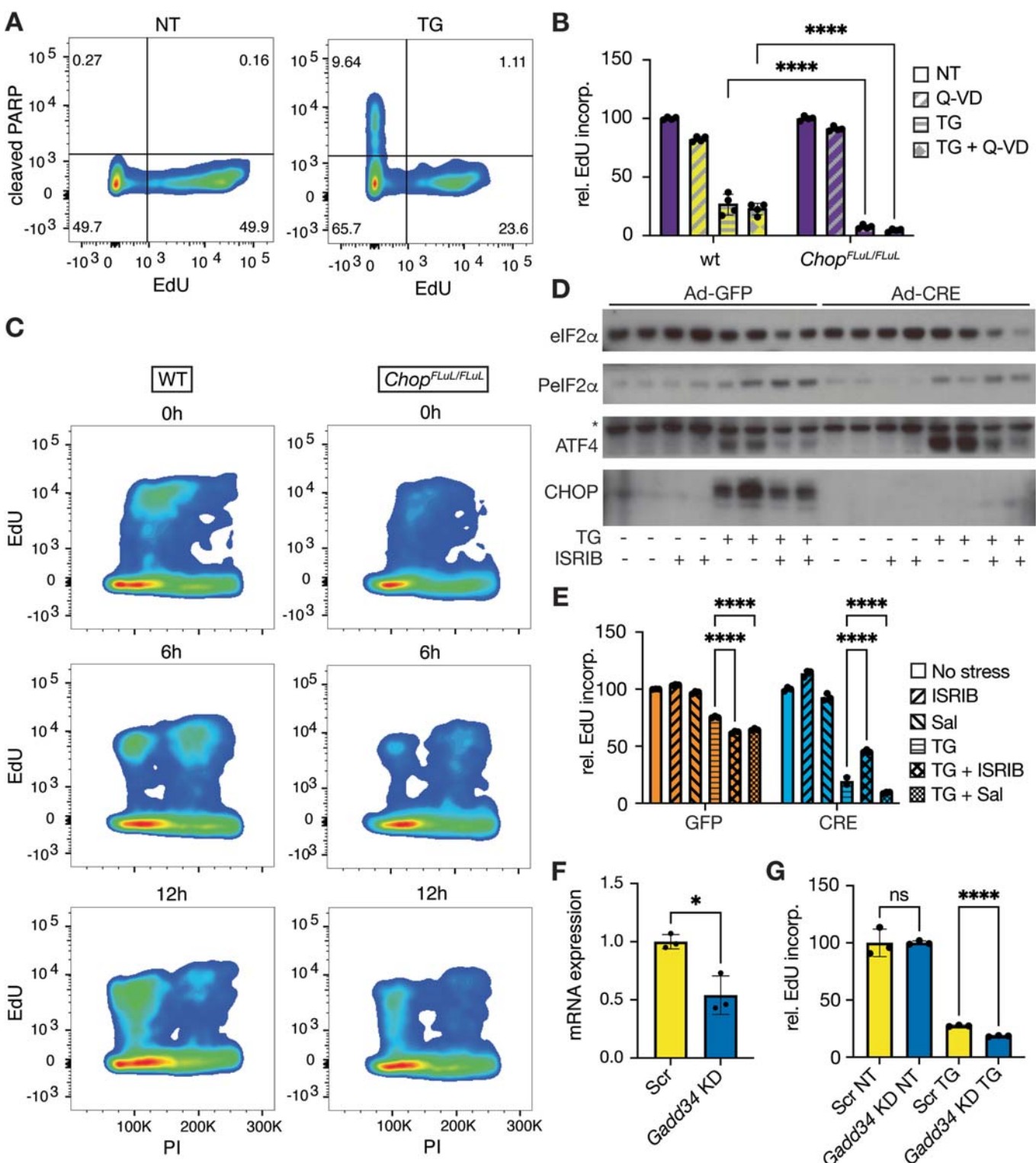

readout for such a complex process. We mixed *Chop^FLuL/FLuL* cells treated with either Ad-GFP (CHOP-null) or Ad-FLPo (CHOP-expressing) and challenged them with 10 nM TG for 16 h (Fig. EV4A–D). We chose this dose because it is the lowest at which we can reliably observe a CHOP-dependent difference in EdU incorporation (Figs. 3C,D and EV4A). The fact that the nLuc

cassette is present only in the *Chop^FLuL/FLuL* cells but not the Chop^fl/fl cells (Fig. 2A,B) and that nLuc is expressed even under non-stressed conditions (Fig. 2D), allowed us to discriminate cells of each genotype within the population by their expression of nLuc or lack thereof. Indeed, while there was a small percentage of cells that express neither *Chop* nor *nLuc* (cluster 6), the expression profiles of

**Figure 4. CHOP-stimulated proliferation occurs independently from cell death.**

(A) Flow cytometry analysis of EdU and cleaved PARP in w.t. primary MEFs after vehicle or 50 nM TG treatment for 24 h. (B) w.t. and *Chop^{FLuL/FLuL}* primary MEFs were treated with vehicle or 50 nM TG, with or without the pan-caspase inhibitor Q-VD-OPh (10 μM) for 24 h. EdU was added 6 h before cell harvest. Statistical comparisons were between genotypes. Means +/− S.D.M. from independently treated wells. Two-way ANOVA with Šidák, ****$p < 0.0001$. (C) w.t. and *Chop^{FLuL/FLuL}* primary MEFs were cultured in 5 mM DTT-containing media for 2 h to induce ER stress. Then the DTT-containing media was changed to fresh, stressor-free media. 15 h later, cells were pulse labeled with EdU for 30 min. Then EdU-containing media was changed to normal culture media. The cell cycle status of EdU-labeled cells was evaluated at 0, 6, and 12 h after the EdU pulse by EdU and PI double staining. (D) Ad-GFP- or Ad-CRE-infected *Chop^{fl/fl}* MEFs were treated with vehicle or 50 nM TG, with or without 1 μM ISRIB for 24 h. The expression of CHOP, ATF4, and total and phosphorylated eIF2α were assessed by western blot. Asterisk for ATF4 blot denotes a non-specific band that also shows loading. (E) Cells were treated as in (D) except also with or without 25 μM Salubrinal (Sal) as indicated, with EdU added 4 h before analysis for EdU positivity. Means +/− S.D.M. from independently treated wells. Two-way ANOVA with Šidák, ****$p < 0.0001$. (F) *Gadd34* expression was assessed by qRT-PCR in MEFs transfected with control or *Gadd34*-targeting siRNA. Means +/− S.D.M. from independently treated wells. Student's t-test. *$p < 0.05$. (G) EdU incorporation was assessed in control or *Gadd34*-knockdown cells after 24 h treatment with vehicle or 50 nM TG. Means +/− S.D.M. from independently treated wells. Welch's ANOVA because data did not satisfy normality criterion. ns = not statistically significant; ****$p < 0.0001$. Source data are available online for this figure.

*Chop* and *nLuc* are otherwise essentially mutually exclusive (Fig. 7A). UMAP (Uniform Manifold Approximation and Projection) analysis separated cells into two major groups of cells, and the Leiden algorithm identified 15 potentially distinct clusters within the population (Fig. 7A; Dataset EV1). Pathway analysis revealed that clusters 0 and 8 comprised proliferating cells, as the "Chromosome Segregation" pathway was dramatically enriched in both groups, and to a much lesser extent, if at all, in any other group (Figs. 7B and EV4E). "DNA-dependent DNA replication" was also enriched in clusters 0 and 8, and also in clusters 6 and 7. Notably, the "Response to ER stress" pathway was not enriched in clusters 0 or 8. The conclusion that clusters 0 and 8 represent proliferating cells was supported by expression of the proliferation markers *Mki67* (Fig. 7C) and *Pcna* (Dataset EV1). Notably, while there are some nLuc-expressing cells (all in group 0), the majority of cells in the two groups are wild-type with respect to *Chop* expression (Fig. 7A,B). Yet, among the groups with substantially expressed *Chop* (1, 2, 0, 8, and 13), *Chop* was lowest in groups 0 and 8 (Fig. 7D). Our pathway results also showed that groups 1, 2, 3, 5, 9, 10, 13, and 14—essentially, the upper right portion of the UMAP plot plus the small group 14—represent cells with an activated UPR at that time point (Fig. 7B), with the UPR (or "response to ER stress") being the most highly upregulated pathway in clusters 2 and 9 (Fig. EV4E). These results support the model that CHOP promotes proliferation, and also suggest that this proliferation occurs in cells with attenuated UPR signaling.

Supporting the *Xbp1* splicing data in Fig. 6, the scRNA-seq data illustrates that CHOP promotes the maintenance of UPR signaling. We arrive at this conclusion based on groups 4 and 7. These two closely related groups comprise predominantly nLuc-expressing (and therefore CHOP-negative) cells (Fig. 7A). UPR activation is lower in these cells than in the clusters in the upper right quadrant of the UMAP graphs, as seen both in lower expression of nLuc (Fig. 7A), the ER cochaperone *Dnajc3*, and other UPR target genes (Fig. 7C and Dataset EV1). (We chose *Dnajc3* to display because, unlike many other genes encoding ER chaperones, it is expressed across a fairly wide dynamic range, making differences in its expression more obvious). A broader analysis of UPR targets genes also supports that groups 4 and 7 are not characterized by UPR activation (Fig. EV4F). Yet these two groups also showed elevated expression of the Integrated Stress Response (ISR) marker genes *Rars*, *Chac1*, *Atf4*, and *Trib3* (Figs. 7C and EV4F). The ISR refers to the gene regulatory pathway that is induced downstream of eIF2α kinases, including PERK but also PKR, HRI, and GCN2 which are

sensitive to other types of stress (Harding et al, 2003). Therefore, while the ISR is one component of the UPR, it can be functionally dissociated from the UPR, as appears to be the case in clusters 4 and 7. From this result, we conclude that loss of CHOP increases the likelihood that a cell will have a gene expression profile consistent with suppression of the UPR but hyperactivation of the ISR. This finding is also consistent with the previously published observation that CHOP restrains ISR activity (Kaspar et al, 2021). We also found when we profiled the expression of a larger range of UPR target genes that a majority of them (12/20) were more highly expressed in the wild-type UPR-activated clusters (1, 2, 13, 14) than in the knockout UPR-activated clusters (4, 7, 9, 10, 11; 7 genes out of 20) (Fig. EV4F). While several of the genes that are more highly expressed in CHOP-positive cells have been identified as direct CHOP targets (*Ero1l*, *Wars*, *Bip*, *Ppp1r15a*) (Han et al, 2013), most have not.

To gain more insight into the functional consequences of CHOP activity, we compared the extent to which individual genes in the dataset correlated with expression of *Chop* versus with *nLuc*, under the assumption that the genes most strongly dependent on CHOP would show the largest difference in correlation between CHOP and nLuc in a way that removed ER stress itself as a confounding variable (since cells with high expression of either CHOP or nLuc would be those cells with strong ISR or UPR activation). Consistent with the conclusions from global RNA-seq data (Han et al, 2013), there was little difference in the expression of a sampling of apoptosis-related genes previously implicated as CHOP targets between cells of the two genotypes, suggesting that they are not strongly affected, if at all, by CHOP under these conditions (Fig. 7E). In contrast, among the 20 UPR target genes shown in Fig. EV4F, 16 of them correlated more strongly with CHOP than with nLuc—some of them substantially so, including previously-identified CHOP targets (*Ppp1r15a/Gadd34*, *Ero1l*, *Trib3*, *Wars*) but also other UPR target genes not previously linked to CHOP or PERK but instead tied to the ATF6 or IRE1 pathways (*Pdia4*, *Pdia6*, *Dnajc3*, *Edem*) (Adamson et al, 2016; Lee Iwakoshi and Glimcher, 2003). This finding that most UPR genes are more strongly upregulated when CHOP is present—including genes that are probably not direct targets of CHOP—is consistent with the idea that CHOP can aggravate ER stress and thus lead to increased expression of some UPR targets that are not themselves directly regulated by CHOP. Interestingly, when we performed pathway analysis of the genes with the strongest correlation difference between CHOP

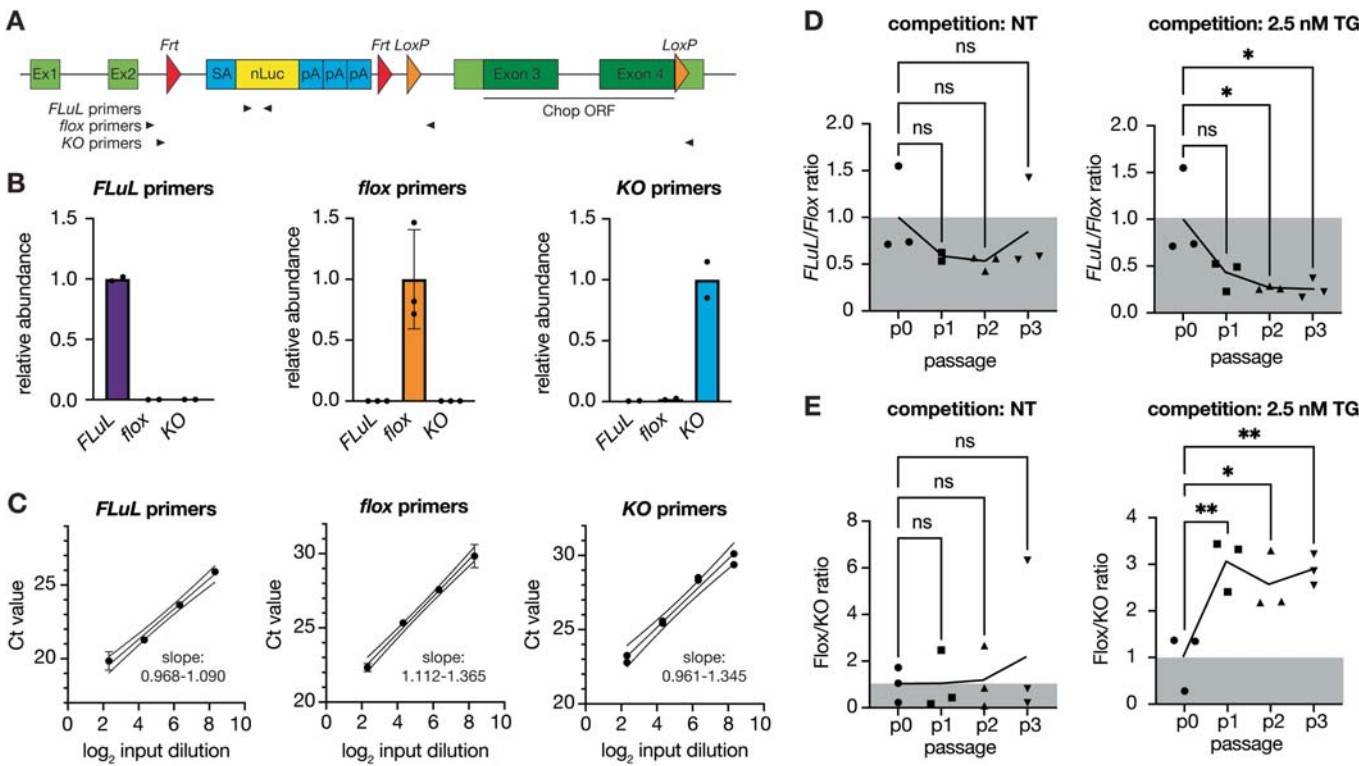

**Figure 5. CHOP confers a competitive proliferative advantage on cells under mild ER stress.**

(A) Graphic depiction of the recognition sites of primer sets (arrowheads) that target the various *Chop* allelic variants. (B) DNAs isolated from cells harboring each of the three *Chop* alleles was mixed in equal amounts, and quantitative PCR using each of the primer pairs shown in (A) was used to discriminate each allele. (C) Amplification efficiency of each primer pair was determined using a 4-fold dilution series of that primer pair's target DNA. (D, E) Ad-GFP- and Ad-GFP-FLPo- infected *Chop^{FLuL/FLuL}* cells (D) or Ad-GFP and Ad-GFP-CRE-infected *Chop^{fl/fl}* (E) primary MEFs were mixed at a 1:1 ratio (p0) and co-cultured in vehicle or 2.5 nM TG-containing media in triplicate. Media and stressor were changed every 48 h. When NT cells reached ~90% confluency, cells in both groups (NT and TG) were passaged to a new plate with the same seeding density, with fresh media and stressor added the next morning. An aliquot of cells was collected at every passage up to passage 3. DNA was isolated and quantitative PCR was used to evaluate the frequency of different alleles relative to the starting cells. Each data point is from an independently passaged culture. One-way ANOVA, ns = not statistically significant, *$p < 0.05$; **$p < 0.01$. Source data are available online for this figure.

and nLuc, we found that pathways of DNA replication and cell cycle control were enriched (Fig. 7F; Dataset EV2), which provides further evidence that a major consequence of CHOP action during moderate ER stress conditions is cell cycle progression and proliferation.

## Discussion

CHOP indisputably contributes to cell death during exposure to severe ER stress, and it appears to fulfill this role by promoting the dephosphorylation of eIF2α and the resumption of protein synthesis. To that canon, our work adds the following: (1) CHOP also stimulates recovery from stress; (2) under mild ER stress conditions, this effect confers a proliferative advantage; (3) this effect likely arises from the same molecular mechanism as that by which CHOP promotes death—restoration of protein synthesis; and (4) CHOP expression is transient during mild stress, and proliferation is favored in cells in which UPR activation has eventually been attenuated.

Based on these data, we can put forth a working model to reconcile the seemingly contradictory death-promoting and

proliferation-promoting functions of CHOP (Fig. 8). We propose that the fundamental role of CHOP in cell physiology is to "stress test" the ER. CHOP is not strictly essential for reversing eIF2α phosphorylation and restoring protein synthesis during stress, as that task can be completed by CReP (Jousse et al, 2003). Rather, we propose that, by accelerating this reversal, the role of CHOP is to maximize UPR activation in cells that have not already overcome the stress. In this paradigm, we can envision two populations of cells at the time CHOP expression (and presumably activity) are at their maximum: cells that have successfully restored ER homeostasis and cells that have not. We suspect that the likelihood that a cell finds itself in one versus the other of these two groups is based on variables such as sensitivity to the stressor applied (for instance expression of the SERCA calcium pump targeted by thapsigargin), basal expression level of ER chaperones, position in the cell cycle, and other factors. In the group of successfully adapted cells, the resumption of protein synthesis is probably unproblematic, and likely advantages those cells as they reenter the cell cycle after stress-mediated interruption (Brewer and Diehl, 2000; Brewer et al, 1999; Lee et al, 2019). Moreover, because those cells have restored ER homeostasis, UPR signaling (and with it CHOP expression, since CHOP is extremely labile at both protein and RNA levels

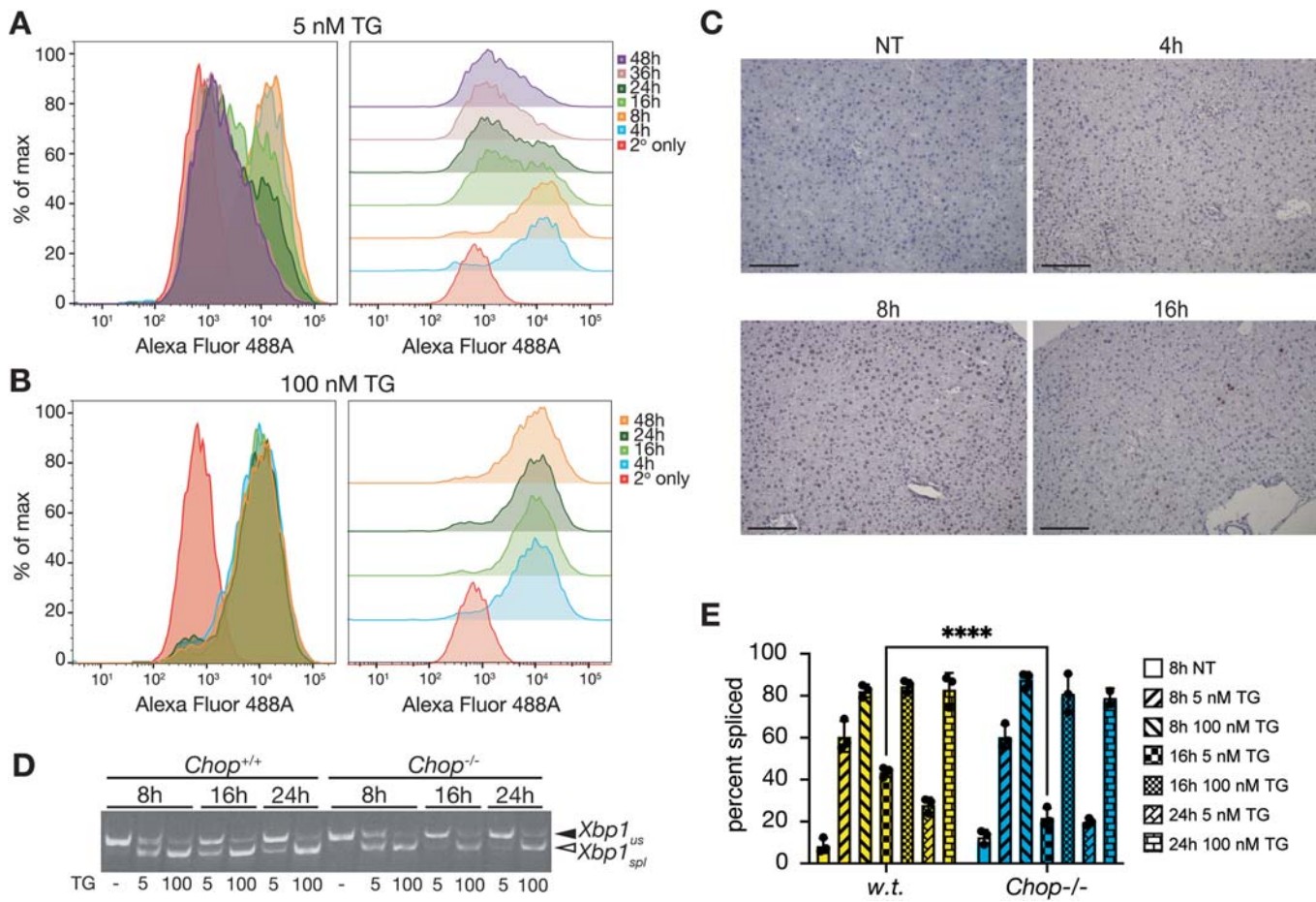

**Figure 6. Mild ER stress splits cells into populations of those expressing and not expressing CHOP.**

(A, B) Wild-type MEFs were treated for the indicated times with 5 or 100 nM TG, after which cells were fixed, permeabilized, and stained to detect endogenous CHOP before analysis by flow cytometry. Experiments were performed together and are separated here for visualization purposes. (C) Wild-type mice were injected intraperitoneally with 1 mg/kg TM, and livers were immunostained for CHOP expression, and counterstained for hematoxylin, at the indicated times after challenge. Scale bar = 50 μm. (D) Wild-type or Chop−/− MEFs treated with vehicle or 5 or 100 nM TG as indicated were harvested at the indicated time points and Xbp1 mRNA splicing was detected by conventional RT-PCR, with the unspliced (us) and spliced (spl) forms indicated. (E) Quantification of Xbp1 mRNA splicing from the experiment shown in (D) conducted in biological triplicate. Means +/− S.D.M. from independently treated wells. Two-way ANOVA with Šidák. ****p < 0.0001. Source data are available online for this figure.

(Rutkowski et al, 2006)) is rapidly attenuated. Conversely, in the latter group—cells that have not restored ER homeostasis—CHOP action can be expected to exacerbate the ER stress burden, in the process prolonging UPR activation (Figs. 6 and 7) and the adaptive measures that flow from that activation such as the IRE1-dependent enhancement of translocon expression and activity (Adamson et al, 2016). We propose that during stresses of typical physiological intensity, this hyperactivation results in death in only a minor population of cells that remain unable to restore ER homeostasis even with the augmented UPR activation. Instead, we speculate that CHOP confers on most of these cells greater odds of then overcoming the stress burden, restoring ER homeostasis, and shutting off UPR signaling (and, with it, CHOP expression; Fig. 8, left).

This model proposes that CHOP effectively drives cells into two populations—those that are fully adapted and primed to resume cell growth and proliferation, and those in which even maximal UPR activation is insufficient to restore homeostasis and which are

eventually doomed to die (Fig. 8, right). Such a function could account for why, as an ER stress stimulus persists, cells largely either fully retain or fully suppress CHOP expression, with few cells in an intermediate state (Fig. 6B,C). In this model, CHOP would be acting as a component of a cellular switch. Switches arise from signaling cascades with two fundamental properties: positive feedback—which reinforces the movement of a switch from on to off and vice-versa—and ultrasensitivity, which defines a group of mechanisms that are capable of converting a linear stimulus into an all-or-none response (Ferrell, 1999). Notably, the axis leading from PERK to GADD34 through CHOP was already known to satisfy the ultrasensitivity condition, because one mechanism for generating ultrasensitivity is the presence of a feed-forward circuit (Ferrell and Ha, 2014). CHOP is part of such a circuit, because, while eIF2α phosphorylation stimulates translation of the upstream regulator of CHOP, ATF4 (Harding et al, 2000), it also stimulates translation of CHOP itself (Palam Baird and Wek, 2011) and of GADD34 (Lee Cevallos and Jan, 2009). Likewise, ATF4 transcriptionally regulates

**Figure 7. scRNA-seq analysis supports a proliferative role for CHOP during stress.**

(A) *Chop*<sup>FLuL/FLuL</sup> cells treated with either Ad-GFP-CRE or Ad-FLPo were mixed 1:1 and treated for 16 h with 10 nM TG. scRNA-seq was performed, and UMAP clustering of the data is shown, with identified clusters (left), expression of *nLuc* (middle), and expression of *Chop* (right). (B) Pathway analysis across all 15 clusters. The GO terms shown are those that were the most significant for each of the 15 groups. P-values were calculated with enrichGO, by Over Representation Analysis and Bonferroni correction for false discovery. (C) Specific expression patterns of *Mki67*, *Dnajc3/p58*<sup>IPK</sup>, *Chac1*, and *Rars* in single cells across all clusters. (D) Violin plots showing *Chop* (left), *Grp94* (middle), and *Bip* (right) expression in different clusters. Cells expressing nLuc (which applies mostly to cluster 0) were removed from the analysis so that in cluster 0 only the wild-type cells (i.e., capable of expressing CHOP) are shown. Significances were calculated by Wilcoxon test. ns = not statistically significant; *$p < 0.05$; ****$p < 0.0001$. (E) The correlations of apoptosis-related genes (left) and UPR target genes (right) with expression of either *Chop* (blue bars) or *nLuc* (orange bars) across the entire dataset are shown. Analysis by Spearman Correlation. (F) Genes showing the greatest difference between correlation with *Chop* and with *nLuc* (r²$_{Chop}$ − r²$_{nLuc}$ > 0.20) were subjected to pathway analysis, with all significant ($p < 0.05$ by Fisher's Exact test and Benjamini-Hochberg correction for false discovery) pathways shown. Source data are available online for this figure.

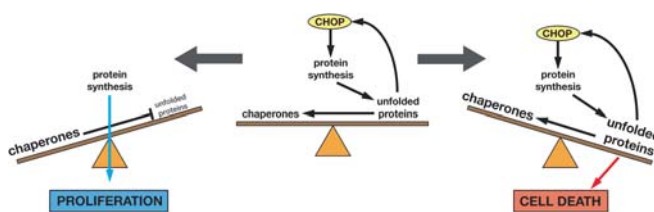

**Figure 8. Model for how CHOP promotes both proliferation and death.**

See Discussion for details.

both CHOP and GADD34 (Harding et al, 2000; Ma and Hendershot, 2003). Ablation of this feed-forward loop in silico pointed to its role in maximizing quicker UPR activation and also quicker resolution (Diedrichs et al, 2018). And, the exacerbation of ER stress by CHOP would complete a positive feedback circuit, because CHOP would then enhance signaling through the three arms of the UPR, and all three of those arms in turn converge on CHOP expression (Acosta-Alvear et al, 2007; Donati Imbriano and Mantovani, 2006; Ma et al, 2002). We suspect that such a role for CHOP has largely escaped notice because the typical ER stress conditions applied to cells experimentally are severe enough that few or no cells are capable of actually adapting to them.

While we believe this model best accounts for our data, we also acknowledge that it awaits further testing. The effect of CHOP in

perpetuating UPR signaling could be brought about by CHOP prolonging ER stress, as we propose, but it could also arise from CHOP regulating UPR output independent of ER stress. We favor the former model because the effect of CHOP on UPR genes does not appear limited to those which have already been placed downstream of CHOP. These two models differ in their predictions about how the ER proteostasis environment is affected by CHOP, with only the former model predicting that CHOP will transiently exacerbate ER protein misfolding. We were limited to using UPR activation as a readout for ER stress, but ultimately the ability to sensitively examine ER proteostasis directly in single cells expressing or lacking CHOP will be needed to discriminate between these models.

In addition, we hope to eventually be able to sort CHOP-expressing and non-expressing cells to ask how these populations differ from each other. However, the high basal expression of nLuc, even though it is knocked into the endogenous *Chop* locus and likely subject to the same transcriptional and translational control as CHOP itself, points to the technical difficulties inherent in creating a fluorescently modified version of CHOP that behaves identically to the endogenous protein. scRNA-seq was a first, albeit imperfect, approach to assessing how UPR activation in individual cells is affected by CHOP. At the same time, the approach yielded unexpected insights about how heterogenous the response of the cells was to a nominally uniform exposure to the stressor. The non-identical expression patterns of *Bip/Grp78/Hspa5* and *Grp94/Hspb90b1*, despite the fact that these two genes are thought to be regulated similarly (Shen et al, 2002; Yoshida et al, 1998), illustrates that, even among the groups in which the UPR is most highly activated (principally the upper right quadrant of the UMAP plot), there exist separable profiles of UPR activation. Likewise, the broader analysis of UPR target genes in Fig. EV4F shows a wide array of gene expression profiles that defies assortment into simple ATF6-, PERK-, and IRE1-dependent groups. Given the extensive overlap among UPR pathways—for example the coregulation of genes with ERSEs by ATF6 and XBP1 (Wu et al, 2007; Yamamoto et al, 2007) and the large influence of PERK on the expression of ATF6- and IRE1-dependent genes (Teske et al, 2011; Wu et al, 2007), this heterogeneity is perhaps unsurprising.

The largely CHOP-negative groups on the left half of the UMAP plot—particularly groups 4, 7, and 10—were best characterized not by UPR activation but by upregulation of rRNA processing and ribonucleoprotein complex biogenesis pathways (Fig. 7B). These functions have been proposed to be independent of the ISR (Bugallo et al, 2020), but we note that the representative ISR target genes *Chac1* and *Rars* were also upregulated in these groups. The functional consequences of this partitioning are unclear, but suggest that loss of CHOP promotes the population of a completely distinct group of cells experiencing stress but with a muted UPR and upregulated ribosomal biogenesis. While seemingly at odds with the idea that CHOP promotes protein synthesis, it is consistent with our previous findings in vivo that loss of CHOP leads to upregulation of genes encoding ribosomal proteins (DeZwaan-McCabe et al, 2013). It could be that CHOP promotes short-term enhancement of the activity of existing ribosomes by upregulating tRNA synthetases, promoting dephosphorylation of eIF2α, etc., while simultaneously restraining translation longer-term by suppressing ribosome biogenesis. We finally note that

group 6 appears to comprise cells with no apparent activation of either the ISR or the UPR. It is not yet clear whether these cells were completely refractory to ER stress, or instead whether they resolved it completely. Together, these data highlight the value of exploring UPR dynamics at the single cell level to truly understand how its output is linked to cell fate.

A major question moving forward is if, and if so how, the function ascribed to CHOP here impacts either organismal physiology or disease. Here, the readout for CHOP's function was proliferation, and we observed a CHOP dependence in MEFs and an immortalized liver cell line, both of which are proliferative. While some cell types remain highly proliferative in vivo—notably the stem cells that replenish the blood, endothelium, skin, and digestive tract epithelia—most cells in adult animals quiesce after differentiation, retaining in some cases the capacity to proliferate given the proper tissue injury cues (for example cells such as skin fibroblasts or hepatocytes) (Goodell Nguyen and Shroyer, 2015; Krizhanovsky et al, 2008). Because such injury cues often have an ER stress component (Huang Xie and Liu, 2014; Mollica et al, 2011), it is possible that CHOP serves a dual death- and proliferation-promoting role in such circumstances as well. Most published studies examining physiological or pathophysiological roles for CHOP (including our own prior studies) have done so using mice with a constitutive deletion of the gene, which raises the possibility of compensatory adaptations that obscure the true function of the protein. It is also possible that increased proliferation is, as a readout for recovery from stress, somewhat unique to cells that proliferate robustly in the absence of stress. And even in such cells, it is not yet clear whether the proliferative advantage conferred by CHOP persists or instead whether wild-type cells eventually lose this advantage. In other less proliferative or non-proliferative cell types, the benefits of CHOP might be realized in other ways. By effectively hastening adaptation to stress, CHOP might be expected to restore normal cellular activities even in non-dividing cells—for example action potentials or neuro-transmission in neurons, metabolic control in hepatocytes, and so on. As in vitro, such benefits might only be observed during stresses of mild intensity, which most experimental challenges, such as injection of mice with tunicamycin, are not. And, as in vitro, there also remains the challenge of separating other, perhaps beneficial outcomes of CHOP action from its clear role in cell death. Just as it is true in vitro that CHOP simultaneously promotes both adaptation (as realized by proliferation) and death, it might have discrepant effects on individual cells within a population in vivo as well. We do note that, while loss of CHOP often correlates with diminished cell death and/or phenotypic improvement in the various experimental models of disease in which its function has been tested (for example (Fang et al, 2023; Park et al, 2022; Song et al, 2008; Uzi et al, 2013)), it has not to our knowledge ever been definitively demonstrated that the contribution of CHOP to any given disease model is *caused by* cell death. In fact, in several cases it has been suggested that CHOP's effects on cells are mediated not through cell death but through other pathways, most notably including inhibition of differentiation (Batchvarova Wang and Ron, 1995; Pennuto et al, 2008). Moreover, deletion of CHOP is not always protective in ER stress-linked disease models (for example (Campos et al, 2014; Germani et al, 2022; Grant et al, 2014; Gurlo et al, 2016; Nemeth et al, 2022)), raising the possibility that at least in some cases it has solely a protective role. Ultimately, we hope

that the unique allele created here, which allows CHOP to be either deleted or restored in a tissue-specific fashion, will best allow these questions to be addressed.

From this work, we propose a reconsideration of the widely-held idea that the role of CHOP is to promote cell death during ER stress, in favor of the view that its more precise role is to drive cells into either adaptation or death. We hope that our findings stimulate the development of better approaches for probing the consequences of mild ER stress, to individual cells, in vivo.

# Methods

## Generation of the Chop FLuL and Chop Flox allele

$Chop^{FLuL/FLuL}$ mice were generated by CRISPR/Cas9-mediated targeting, carried out by the University of Iowa Genome Editing Facility, using techniques based on (Miura et al, 2018). C57BL/6J mice were purchased from Jackson Labs (000664; Bar Harbor, ME). Male mice older than 8 weeks were used to breed with 3–5-week-old super-ovulated females to produce zygotes for pronuclear injection. Female ICR (Envigo; Hsc:ICR(CD-1)) mice were used as recipients for embryo transfer. All animals were maintained in a climate-controlled environment at 25 °C and a 12/12 light/dark cycle.

Chemically modified CRISPR-Cas9 crRNAs and CRISPR-Cas9 tracrRNA were purchased from IDT (Alt-R® CRISPR-Cas9 crRNA; Alt-R® CRISPR-Cas9 tracrRNA (Cat# 1072532)). The crRNAs and tracrRNA were suspended in T10E0.1 and combined to 1 µg/µl (~29.5 µM) final concentration in a 1:2 (µg:µg) ratio. The RNAs were heated at 98 °C for 2 min and allowed to cool slowly to 20 °C in a thermal cycler. The annealed cr:tracrRNAs were aliquoted to single-use tubes and stored at −80 °C.

Cas9 nuclease was also purchased from IDT (Alt-R® S.p. HiFi Cas9 Nuclease). Cr:tracr:Cas9 ribonucleoprotein complexes were made by combining Cas9 protein and cr:tracrRNA in T10E0.1 (final concentrations: 300 ng/µl (~1.9 µM) Cas9 protein and 200 ng/µl (~5.9 µM) cr:tracrRNA). The Cas9 protein and annealed RNAs were incubated at 37 °C for 10 min. The RNP complexes were combined with single-stranded repair template and incubated an additional 5 min at 37 °C. The concentrations in the injection mix were 100 ng/µl (~0.6 µM) Cas9 protein and 20 ng/µl (~0.6 µM) each cr:tracrRNA and 40 ng/µl single-stranded repair template.

Pronuclear-stage embryos were collected in KSOM media (Millipore; MR101D) and washed 3 times to remove cumulous cells. Cas9 RNPs and double-stranded repair template were injected into the pronuclei of the collected zygotes and incubated in KSOM with amino acids at 37 °C under 5% $CO_2$ until all zygotes were injected. Fifteen to 25 embryos were immediately implanted into the oviducts of pseudo-pregnant ICR females. Guide RNA sequences were as follows:

Ddit3_5PACTAATGATGGTGTGTCGGGA
Ddit3_3PCCTGCACCAAGCATGAACAGT

Correct targeting was verified by sequencing through the allele in founder mice. $Chop^{fl/fl}$ mice were generated by breeding $Chop^{FLuL/+}$ mice with the FLP delete strain $Pgk1\text{-}flpo$ in the C57BL/6J background (Jackson labs strain 011065; (Wu et al, 2009)) and then breeding the allele to homozygosity. All animal

usage was approved by and in accordance with Institutional Animal Care and Use Committee procedures Protocol #3021076.

Primers used for genotyping were:
Primer A: TGGATCTGGCAGGGTCAAAG
Primer B: CCCAACCCCTTCCTCCTAC
Primer C: TGGAAAGGACATACATTCCA

With these primers, the w.t. product is 270 bp, the $FLuL$ product is 194 bp, and the $Flox$ product is 382 bp.

## Cell Culture and drug treatments

Primary MEFs of different genotypes were isolated by timed intercrosses of $Chop^{+/-}$ (Zinszner et al, 1998), $Chop^{FLuL/+}$, or $Chop^{fl/+}$ animals. MEFs of the various genotype combinations from the $FLuL$ allele are available upon request. All MEF experiments were repeated in at least two independently-derived cell lines to confirm robustness of data. Female mice were euthanized at 13.5 days post-coitus and MEFs were isolated following the procedure described by Marian et al, 2013. Genotypes of isolated MEFs were determined by PCR. For cell culture, MEFs were maintained in DMEM containing 4.5 g/L glucose (Invitrogen, USA) supplemented with 10% FBS, 1% L-glutamine, penn/strep, amnio acids, and non-essential amino acids (Invitrogen, USA), at 37 °C in a 5% $CO_2$ incubator. Cell cycle synchronization was carried out by culture for 48 h in serum-free medium prior to stress treatments in complete medium. For chemical treatment, 1000X stocks of TG (EMD/Millipore), TM (EMD/Millipore), ISRIB (EMD/Millipore), Q-DV-OPh (Cayman), ISRIB (EMD/Millipore), and Salubrinal (EMD/Millipore) were made in DMSO and 200X stocks of DTT were made in 1x PBS, and single-use aliquots were frozen. For non-treated cells, DMSO or PBS was added to the same concentration as for treated cells. The plating densities of cells were $2.0–2.5 \times 10^5$ cells/well in 12-well plates (for pulse labeling assay), $3.0–4.0 \times 10^5$ cells/well in six-well plates (for protein and RNA analysis), $7.0–8.0 \times 10^5$ cells/well in 60 mm dishes (for flow experiments), and cells were allowed to rest overnight before stressor treatment. For growth competition experiments, the media was removed and replaced with fresh media containing stressor every 48 h. Both treated and non-treated groups were passaged when non-treated cells reached ~90% confluency. During passaging, both groups were plated at the same density, rested overnight, and then treated again with DMSO or stressor-containing media the next day. For Q-DV-OPh, the stock concentration of the drug was 10 mM (in DMSO). Cell cultures were periodically confirmed to be mycoplasma-negative. In vivo challenge with TM was performed intraperitoneally with TM or DMSO diluted in PBS.

## Other cell culture

To delete CHOP in TIB-73 cells (RRID:CVCL_4383), gRNAs targeting the N-terminus of CHOP were cloned into the pX458 plasmid backbone that also expresses Cas9 and GFP separated by a P2A cleavage site (Ran et al, 2013). TIB-73 cells were transfected with either empty vector or gRNA-containing vector using Lipofectamine 3000 (Thermofisher, USA). Forty-eight hours after transfection, GFP positive cells (from both groups) were sorted by flow cytometry and plated on 96-well plates (1 cell per well). After obtaining single cell-derived colonies, cells were trypsinized and expanded, and screened for CHOP expression by treatment with

TM (5 µg/ml for 24 h) followed by western blot. For GADD34 knockdown, MEFs were seeded on 60 mm cell culture dish at $0.7 \times 10^6$ cells/dish. Cells were allowed to attach overnight before siRNA transfection. siRNA transfection was performed following manufacture's protocol (MEF Avalanche Transfection Reagent, EZ biosystems, EZT-MEFS-1) using either scrambled negative control siRNA (IDT, 51-01-14-04) or validated GADD34 siRNA (IDT, mm.Ri.Ppp1r15a.13.2). Cells were treated with TG 24 h after transfection. For LDH assay, cells were treated with vehicle or TG for 24 h and cytotoxicity was assessed using the CyQuant LDH Assay (ThermoFisher) according to the manufacturer's protocol.

## Flow cytometry and immunostaining

EdU (final conc 10 µM) was added directly to the culture medium 4 h before cell harvest unless specified elsewhere. Upon harvest, cells were trypsinized and resuspended in 1X PBS for EdU staining. EdU staining was performed using a Click-iT EdU Alexa Fluor 488/594/647 Flow Cytometry Kit (Thermofisher, USA). After staining, cells were resuspended in 1X PBS and analyzed using an LSRII flow cytometer (Becton Dickinson). For PARP staining, after EdU staining cells were resuspended in incubation buffer (0.5% BSA in 1x PBS) in the dark at RT for 30 min. After incubation, cells were then resuspended in antibody staining solution (cleaved-PARP antibody (1:100 dilution, Cell Signaling Technology, 94885) in incubation buffer) and incubated at RT for 1.5 h. Cells were then washed three times with 1x PBS and incubated in secondary antibody staining solution (1:500 dilution, goat anti-rabbit Alexa488 (Thermofisher A-11008)) at RT for 1 h. Cells were then washed four times in 1x PBS and subjected to flow cytometry. For cell cycle analysis, wild-type and $Chop^{FLuL/FLuL}$ primary MEFs were cultured in 5 mM DTT-containing media for 2 h to induce ER stress. The DTT- media was then changed to fresh, stressor-free media after washing with 1x PBS. Fifteen hours later, EdU was directly added to culture medium at a final concentration of 10 µM. After a further 30 min, the EdU-containing media was changed to normal culture media after washing in 1x PBS. The cell cycle status of EdU-labeled cells was evaluated at 0, 6, and 12 h post-EdU pulse. EdU staining was as above. After EdU staining, cells were resuspended in PI staining solution (100 µg/RNase, 20 µg/mL PI in 1x PBS) and incubated in the dark at 37 °C for 30 min. Samples were analyzed on a Becton Dickinson LSRII immediately after PI staining. Data was analyzed using FlowJoV10. For flow experiments, SSC-FSC voltage was set based on pilot experiments on the same cell type. We first gated on the FSC-A-SSC channel to gate for the main cell population, and then doublets were excluded by an additional gate on the FSC-W channel. Proper voltage for fluorescent quantification (e.g. EdU, cleaved-PARP, CHOP) were set based on the results from negative controls (secondary antibody-only). For cell cycle analysis, PI gating was set to only include cells that contain 2N and 4N, excluding cells with polyploidy.

For CHOP immunostaining in MEFs, cells were fixed in 4% formaldehyde for 10 min at RT and permeabilized with 0.1% Triton X-100 (in 1x PBS) for 20 min at RT. After permeabilization, cells were washed twice with 1x PBS, resuspended in 245 µL incubation buffer (10% BSA in 1x PBS), and incubated at RT for 30 min. After blocking, CHOP antibody (Santa Cruz, sc-7351) was added at a 1:50 dilution, and cells were then incubated at RT for 1 h for primary incubation. After primary incubation, cells were washed twice with incubation buffer and twice with 1x PBS and then resuspended with Alexa-488-conjugated anti-mouse antibody (Invitrogen, A21202) diluted 1/500 in incubation buffer at RT in the dark for 1 h. Cells were then washed three times with 1x PBS and analyzed by flow cytometry. Detection of CHOP in fixed liver tissue was as described (DeZwaan-McCabe et al, 2013), following IP injection of 1 mg/kg TM or vehicle diluted in PBS.

## Isolation of paired MEFs lines

At passage 2, $Chop^{FLuL/FLuL}$ or $Chop^{fl/fl}$ MEF (at least two lines of each genotype) were infected with either Ad-GFP or Ad-GFP-FLPo (for $Chop^{FLuL/FLuL}$ MEFs) or Ad-GFP-CRE (for $Chop^{fl/fl}$ MEFs) at an M.O.I. of 250. Adenoviruses were prepared by the University of Iowa Viral Vector Core using the RAPAd system (Anderson et al, 2000). Seventy-two hours after infection, MEFs were trypsinized and resuspended in 1x PBS. Viable GFP-positive MEFs were sorted by FACS (Becton Dickinson Aria II) after Hoechst 334342 staining and FACS. $5 \times 10^5$ cells of each pair were sorted out from each group, cultured on cell culture dishes and expanded for future use. All experiments were performed using cells below passage 10. Matched cell lines are available upon request.

## RNA and protein analysis

Protein lysates were processed for immunoblots as described (Rutkowski et al, 2006). Primary antibodies include: CHOP (Santa Cruz sc-7351 RRID:AB_627411 for flow cytometry or Proteintech 15204-1-AP RRID:AB_2292610 for western blot), PARP (Cell Signaling Technology 9542 RRID:AB_2160739), ATF4 (Cell Signaling Technology 11815 RRID:AB_2616025), PeIF2α (Thermo 44-728G), total eIF2α (Cell Signaling Technology 9722 RRID:AB_2230924), and Calnexin (loading control; Enzo ADI-SPA-865 RRID:AB_10618434). Nanoluciferase activity was assessed using the Nano-Glo In-Gel Detection kit (Promega, USA) following the manufacturer's protocol. qRT-PCR, including primer validation by standard curve and melt curve analysis, was also as described (Rutkowski et al, 2006). Briefly, RNA was isolated using Trizol (Thermofisher, USA) following the manufacturer's protocol, and RNA concentrations were evaluated using the Qubit RNA Broad Range kit (Invitrogen, USA). 400 ng of RNA was used for reverse transcription using PrimeScript RT Master Mix (Takara, USA). PCR reactions were performed using TB Green Premix Ex Taq (Takara, USA). Gene expression was normalized against the average of two loading controls (Btf3 and Ppia).

## qRT-PCR Primers

Btf3 forward: CCAGTTACAAGAAAGGCTGCT reverse: CTTCACACAGCTTGTCCGCT

Ppia forward: AGCACTGGAGAGAAAGGATT reverse: ATTATGGCGTGTAAAGTCACCA

FLuL/Chop exon 1-2 forward: TTGAAGATGAGCGGGTGGCA reverse: CTTTCAGGTGTGGTGGTGTA

FLuL exon 2-nLuc forward: GTGTTCCAGAAGGAAGTGCA reverse: CTTTGGATCGGAGTTACGGA

Chop exon 3-4 forward: GGAAGCCTGGTATGAGGAT reverse: CCACTCTGTTTCCGTTTCCT

*Gadd34* forward: GAGATTCCTCTAAAAGCTCGG reverse: CAGGGACCTCGACGGCAGC

## ³⁵S pulse labeling

For protein synthesis experiments, cells were labeled using an ³⁵S Met/Cys labeling mixture at 200 μCi/ml (EasyTag Express³⁵S; Perkin Elmer) for 30 min in media that contained drug treatments and that was 20 percent MEF culture medium as above and 80 percent DMEM lacking Met/Cys and with dialyzed FBS. After labeling, cells were lysed in lysate buffer (1% SDS, 100 mM Tris (pH 8.9)) followed by vigorous boiling. Samples were separated by electrophoresis, the gels were dried using a vacuum dryer, and then exposed to film at −80 °C (BioMax MS with a Transcreen LE intensifying screen).

## Media switch assay

Wild-type and *Chop^{FLuL/FLuL}* primary MEFs (two lines of each genotype) were treated with vehicle or 50 nM TG for 20 h (original cells). Media switch was performed by collecting media from each group of these original cells and applying the collected media to a separate set of non-treated w.t. primary MEFs (media-receiving cells). The original cells were washed three times with 1X PBS and maintained in fresh, stressor-free media. After media switch, EdU was added directly to both original cells and media-receiving cells, and cells were harvested 4 h after the addition of EdU. The data shown is aggregated from 2 lines (duplicates in each line).

## Growth competition assay

Genomic DNA was isolated using the Qiagen Puregene Cell Kit (158043). An extra RNase treatment step was added to avoid the potential contamination of RNA. DNA concentrations from different groups were then quantified using Qubit dsDNA Broad Range kit (Invitrogen, USA) and diluted to the same concentration (10 ng/μL) for PCR. Extension time in the PCR program was 15 s to preclude amplification of long products. The sequences of different primers are listed below. Allele enrichment was normalized to the *Hprt* locus.

*Hprt* forward: CTGCCTCTGCCTCCTAAATG reverse: TGTCGTCTCCCAGAGGATTC

*FLuL* forward: CAACCTGGACCAAGTCCTT reverse: ATTTGGTCGCCGCTCAGAC

*Flox* forward: CATGTGATCATCTGGACAAC reverse: GATGGTGTGTCGGCCACAC

KO forward: AGAGTTGGATCTGGCAGGGT reverse: GAGAGACAGACAGGAGGTGA

## Single-cell RNA sequencing

FACS sorted Ad-GFP- and Ad-FLPo-treated *Chop^{FLuL/FLuL}* MEFs were allowed to grow for at least two passages and the Ad-FLPo-treated cells verified for restoration of CHOP expression and loss of nLuc activity before use. After quality control, GFP and FLPo cells were trypsinized and the cell concentration was determined using a Countess Automated Cell Counter (Thermofisher, USA). The final concentration was the average of three reads from 3 different aliquots of cell suspension. Cells of different genotypes were then mixed at a 1:1 ratio (total number) and plated on a 10 cm plate and allowed to rest overnight before treatment with either 10 nM TG or vehicle. Upon harvest, cells were trypsinized and centrifuged for 3 min at RT. The pellet was then resuspended in HBSS using large orifice tips. The cell suspension was filtered using a 70 μM cell strainer to eliminate aggregates. Cell viability was determined by Countess after Trypan Blue staining. Only samples with over 90% viability proceeded to the next step. Single-cell RNA-sequencing was performed by the Iowa Institute of Human Genetics using the 10x Genomics Chromium Single-Cell System and a Chromium Next GEM Single Cell 3' GEM Library&Gel Bead Kit (10xGenomics, USA), and a total of 5000 cells were sequenced.

For data analysis, nanoluciferase sequence was added to the *Mus musculus* 10 (mm10) to build a new reference genome. scRNA-seq data were then aligned to reference genome by Cellranger (v. 4.0.0) count. The filtered expression matrix with cell barcodes and gene names was further loaded with the 'Read10X' function of the Seurat (v.4.0.0) R package. For quality control, cells with the number of detected genes (nFeature_RNA) <= 250 and mitochondrial content >5% were removed. Next, the Seurat pipeline was used for data normalization, selection of highly variable feature genes, and clustering with default parameters in most cases. Specially, the UMAP algorithm was conducted for dimensionality reduction with top 30 principal components and a resolution of 0.8 was used for 'FindClusters' function. We also used the python Scanpy package to perform clustering analysis and obtained similar results with Seurat. To identify differentially expressed genes for each cluster, 'FindMarkers' function (only.pos = TRUE, min.pct = 0.25, logfc.threshold = 0.25) was used. We further used the 'enrichGO' function in the ClusterProfiler package to conduct GO analysis. Bonferroni correction was used for the adjustment of $p$ value for enrichment analysis. In addition, the significance level for gene expression between different groups was determined by Wilcoxon test. The correlation coefficient of gene expression was calculated by the Spearman Correlation method.

## Statistical analysis

Continuous variables were reported as the mean ± standard deviation. For comparisons of multiple groups, ANOVA was used with correction (Tukey or Šídák) for multiple hypothesis testing, using GraphPad Prism. For Figs. 1B, 4G, a Mann–Whitney test or a Welch's ANOVA, respectively, were used because of non-normality of residuals. A post-correction alpha of 0.05 was used to determine statistical significance. Data points represent biological replicates, usually from one experiment (independently treated and analyzed wells) where the experiment was also repeated independently at least once, and sometimes, where noted, from multiple experiments conducted similarly. For most experiments, 3–4 samples were used. Blinding was not performed. No data points/samples were excluded.

# Data availability

The scRNA-seq dataset produced in this study is available in NCBI GEO with the accession number GSE242707.

## Peer review information

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

## Acknowledgements

Transgenic mice were generated at the University of Iowa Genome Editing Core Facility, supported in part by grants from the NIH and from the Roy J. and Lucille A. Carver College of Medicine. We wish to thank Norma Sinclair, Patricia Yarolem, Joanne Schwarting and Rongbin Guan for their technical expertise in generating transgenic mice. The flow cytometry data were obtained at the University of Iowa Flow Cytometry Facility, which is funded through user fees and the generous financial support of the Carver College of Medicine, Holden Comprehensive Cancer Center, and Iowa City Veteran's Administration Medical Center. The scRNA-seq data were obtained at the Genomics Division of the Iowa Institute of Human Genetics which is supported, in part, by the University of Iowa Carver College of Medicine. In addition, we want to thank Heath Vignes, Michael Shey, and Thomas Kaufman at the University of Iowa Flow Cytometry facility for technical assistance on flow experiments, and thank Mary Boes and Garry Hauser for their advice on scRNA-seq. We also want to thank Anit Shah for his assistance with primary MEF isolation. This work was funded by NIH grant GM115424 to DTR and by funds from the University of Iowa Department of Anatomy and Cell Biology.

## Author contributions

**Kaihua Liu**: Conceptualization; Investigation; Writing—original draft; Writing—review and editing. **Chaoxian Zhao**: Formal analysis. **Reed C Adajar**: Investigation. **Diane DeZwaan-McCabe**: Investigation. **D Thomas Rutkowski**: Conceptualization; Supervision; Funding acquisition; Writing—review and editing.

## Disclosure and competing interests statement

The authors declare no competing interests.

# Expanded View Figures

**Figure EV1. The effects of CHOP on proliferation and cell cycle regulation in MEFs primary and immortalized MEFs.**

(A) FACS gating for the experiment shown in Fig. 1A. (B) Non-synchronized MEFs were treated with vehicle or 50 nM TG for 24 h and assessed for EdU positivity as in Fig. 1B. Means $+/-$ S.D.M. from independently treated wells. Two-way ANOVA with Šidák, ns = not statistically significant, ****$p < 0.0001$. (C) Representative FACS plots of TM data shown in Fig. 1C. (D) MEFs were treated with vehicle or 50 nM TG for 24 h, trypsinized, and resuspended in propidium iodide to assess cell cycle stage by DNA content. Means $+/-$ S.D.M. from independently treated wells. Two-way ANOVA with Tukey, **$p < 0.01$; ***$p < 0.001$. (E) qRT-PCR was used to assess expression of selected cell cycle-related genes in MEFs treated with vehicle or 50 nM TG for 24 h. Means $+/-$ S.D.M. from independently treated wells. Two-way ANOVA with Šidák. Benjamini-Hochberg correction was applied for multiple hypothesis testing. ns = not statistically significant; *$p < 0.05$; **$p < 0.01$. (F) Verification of CHOP knockout in TIB-73 cells (two separate clones each of wild-type and knockout) by immunoblot after treatment with 500 nM TG for 8 h. (G) Immunoblot showing induction of CHOP compared to non-treated cells after 16 h of treatment with the indicated concentrations of TG (prior to the onset of PARP cleavage). (H) Ad-GFP or Ad-CRE infected *Chop$^{fl/fl}$* MEFs (see Fig. 3) were immortalized by transfection with large T antigen followed by low density passaging to select for immortal cells. EdU positivity during ER stress was assessed as in Figure EV1B above. Each data point represents results from an independent clone. Error bars are means $+/-$ S.D.M. Two-way ANOVA with Šidák, ns = not statistically significant, ****$p < 0.0001$. (I) Three separate clones each of control wild-type or CRISPR-deleted CHOP knockout NIH 3T3 cells were assessed for EdU positivity as with TIB-73 cells in Fig. 1D. Means $+/-$ S.D.M. from independently treated wells. Two-way ANOVA with Šidák, ns = not statistically significant.

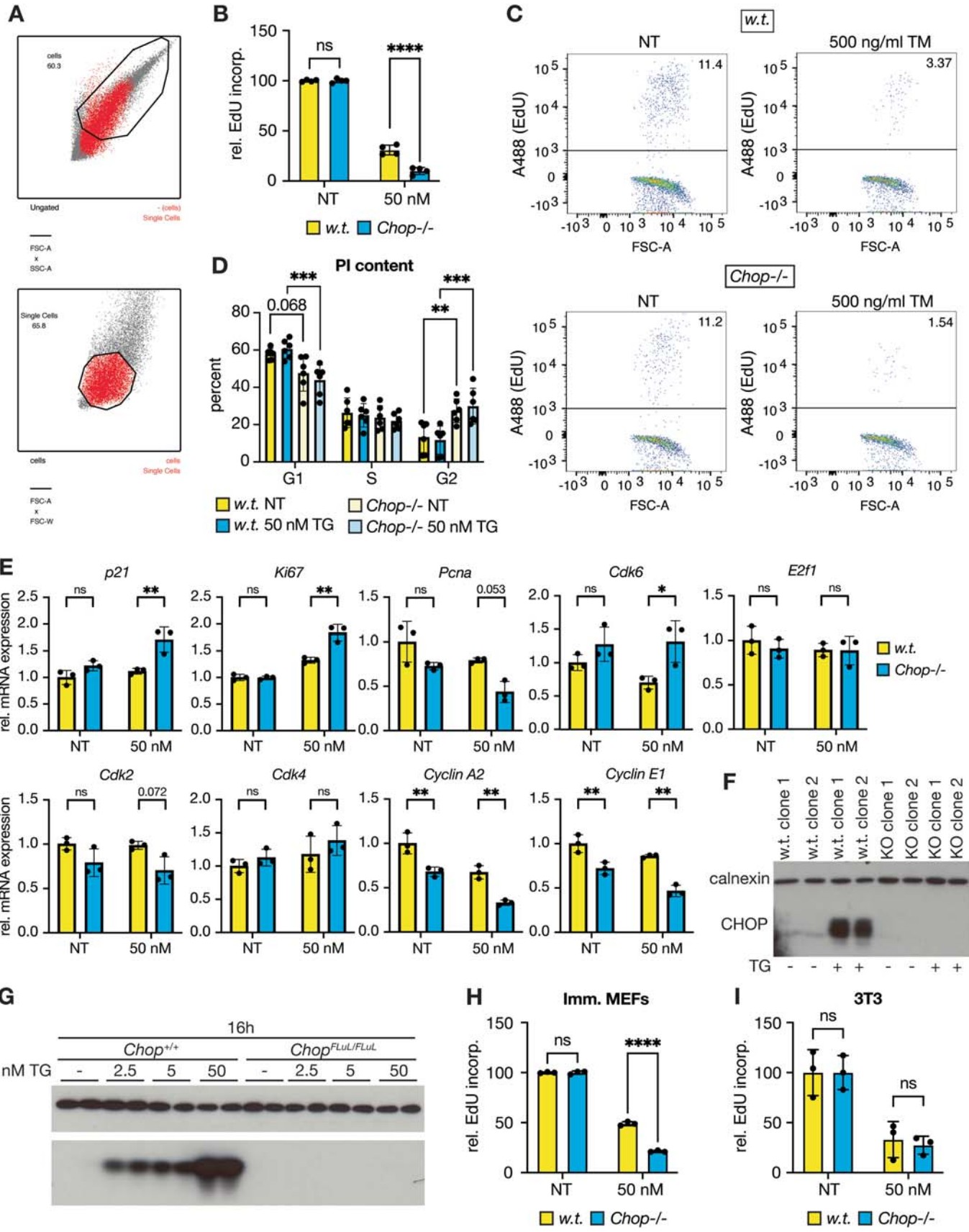

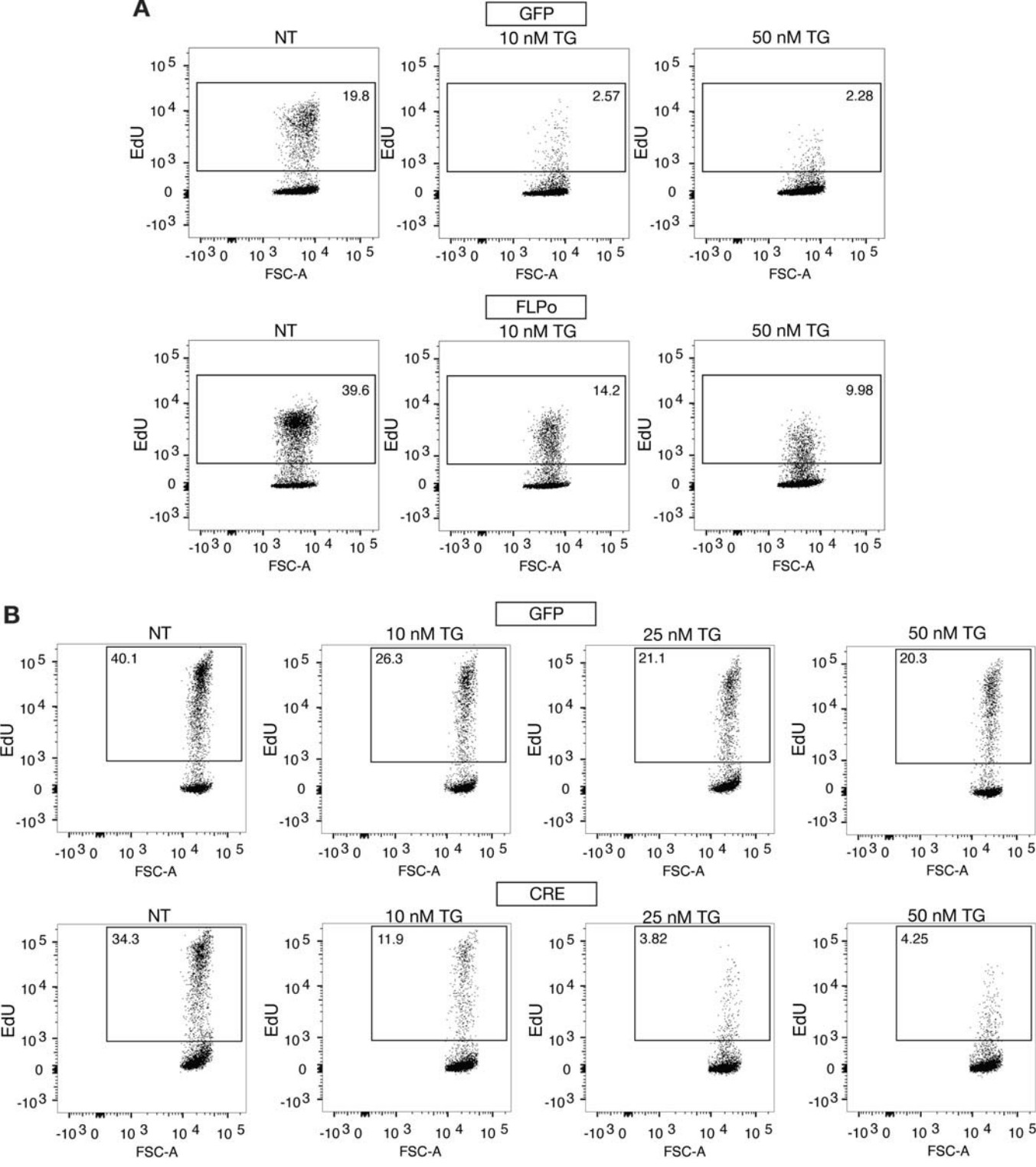

**Figure EV2. Individual flow cytometry plots during TG treatment of CHOP-expressing or non-expressing cells.**

Representative flow cytometry plots from the bar graphs depicted in Fig. 3C (**A**) and 3D (**B**).

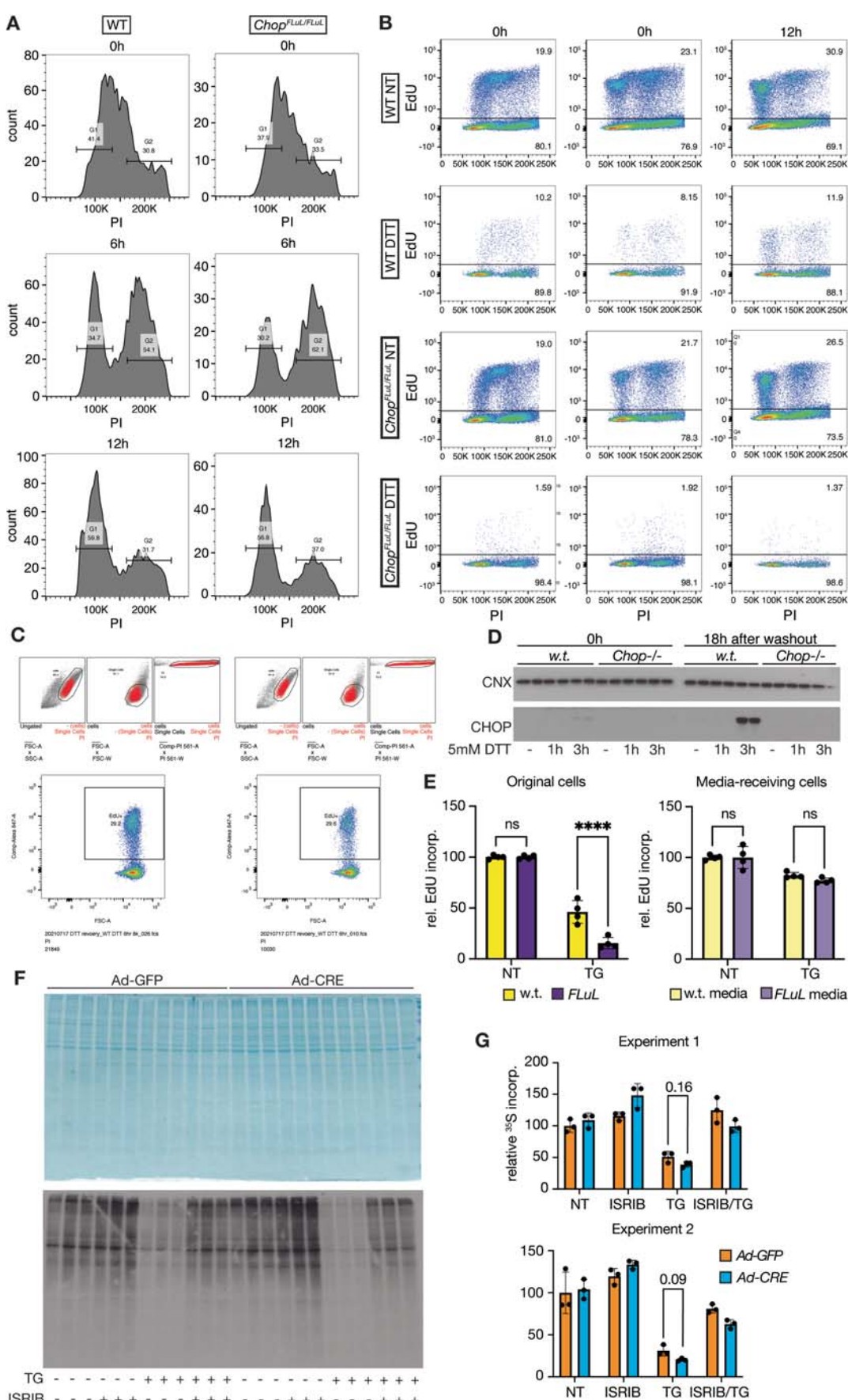

**Figure EV3. Cell cycle progression, cell autonomy of CHOP, and the effects of ISRIB on protein synthesis.**

(A) Histograms showing DNA content from all cells (EdU+ and EdU-) from Fig. 4C. (B) Independent experiment similar to that conducted in Fig. 4C. (C) FACS plots showing gating parameters for cells, single cells, and for PI compensation, respectively (top) and overall measured EdU positivity when different numbers of events were analyzed (>21,000 versus >10,000) (bottom). (D) Immunoblot showing CHOP either immediately after treatment with 5 mM DTT for 1 h or 3 h, or 18 h after DTT washout. (E) *w.t.* and *Chop^{FLuL/FLuL}* primary MEFs were treated with vehicle or 50 nM TG for 20 h (original cells). Media were collected from each group and applied to a separate set of non-treated wild-type primary MEFs (media-receiving cells). At the same time, the original cells were then maintained in fresh, stressor-free media. After the media switch, EdU was added directly to both original cells and media-receiving cells, and the cells were harvested 4 h later. Means +/− S.D.M. from independently treated wells. Two-way ANOVA with Šidák, ns = not statistically significant; ****$p < 0.0001$. (F) Ad-GFP or Ad-CRE-selected MEFs were treated with 50 nM TG and 1 μM ISRIB as indicated. After 4 h of treatment, ^{35}S-methionine/cysteine was added directly to the media at 200 μCi/ml for 30 min. Lysates were analyzed by Coommassie staining (top) and autoradiography (bottom). (G) Quantification of relative labeling efficiency (normalized for total protein load) from two such experiments, conducted independently. Means +/− S.D.M. from independently treated wells. Two-way ANOVA with Šidák with *p*-values shown.

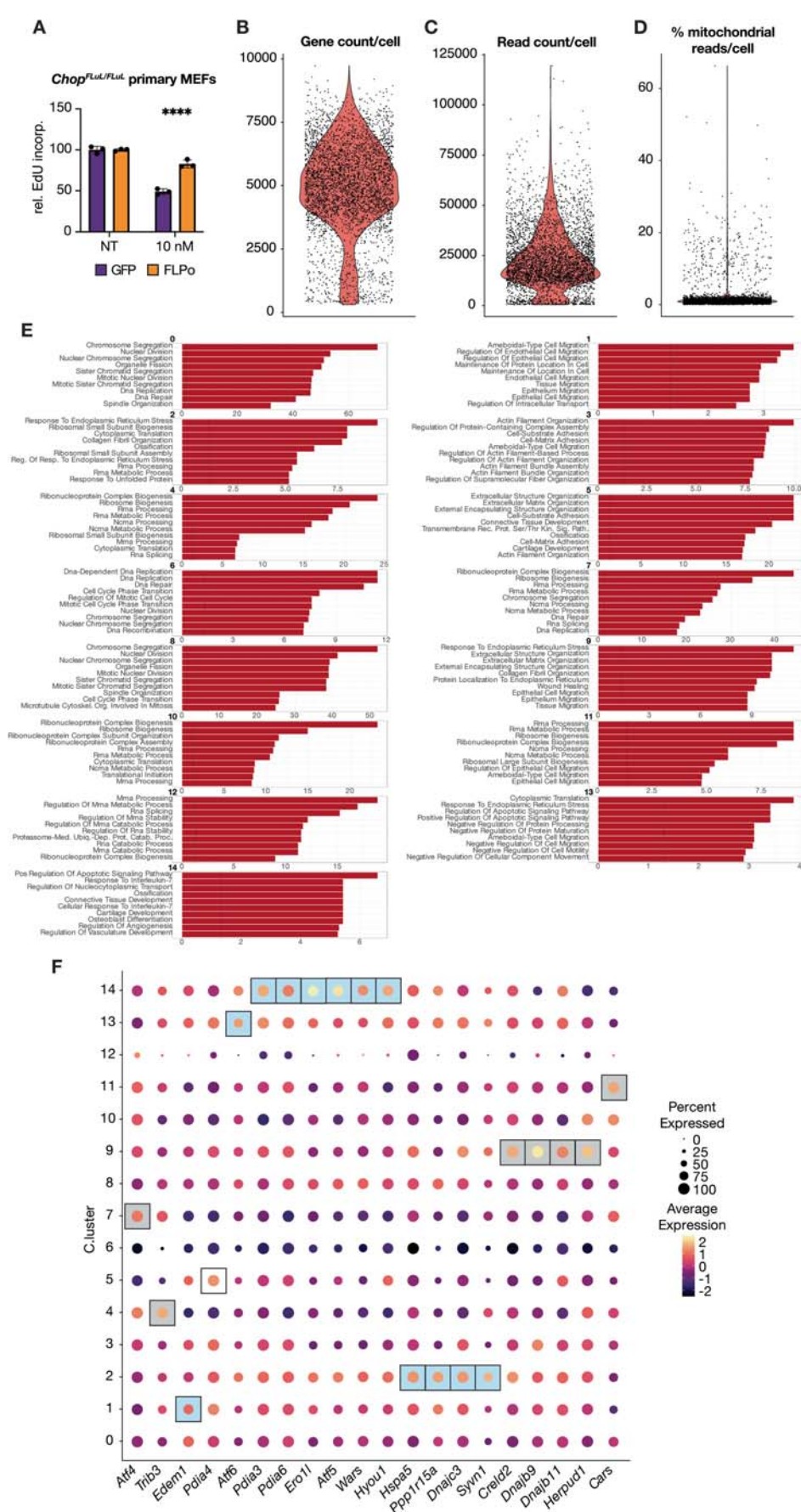

**Figure EV4. Extended scRNA-seq validation and pathway analysis.**

(A) Confirmation of greater EdU positivity in CHOP-expressing (FLPo-treated) cells after treatment with 10 nM TG as in Fig. 7. Means $+/-$ S.D.M. from independently treated wells. Two-way ANOVA with Šidák, ****$p < 0.0001$. (B–D) Violin plots showing gene count, read count, and percentage of reads from mitochondrial transcripts, respectively. (E) Pathway analysis for each of the 15 clusters from scRNA-seq. The x-axis shows the -$\log_{10}$ $p$-value after correction for false discovery. $P$-values were calculated with enrichGO, by over Representation Analysis and Bonferroni correction for false discovery. (F) Relative expression of selected UPR target genes in each cluster is shown. Boxes are used to indicate the cluster in which each gene is most highly expressed, with blue shaded boxes indicating CHOP-positive clusters and gray shaded boxes indicating CHOP-negative clusters (cluster 5 is a mix of both CHOP-positive and -negative cells). Bonferroni correction for false discovery.

