## [Peer Review File · EMBO Reports]

A beneficial adaptive role for CHOP in driving cell fate selection during ER stress

Kaihua Liu, Chaoxian Zhao, Reed Adajar, Diane DeZwaan-McCabe, and D Thomas Rutkowski
DOI: 10.15252/embr.202357249

Corresponding author(s): D Rutkowski (thomas-rutkowski@uiowa.edu)

Review Timeline:

Submission Date:	27th Mar 23
Editorial Decision:	8th May 23
Revision Received:	9th Sep 23
Editorial Decision:	11th Nov 23
Revision Received:	21st Nov 23
Accepted:	23rd Nov 23

Editor: *Martina Rembold*

Transaction Report:

Dear Dr. Rutkowski

Thank you for the submission of your research manuscript to our journal. We have now received the full set of referee reports that is copied below.

As you will see, all three referees acknowledge that your findings are interesting and that the study is overall well executed, but they also raise a number of partially overlapping concerns, e.g., regarding CHOP's role in proliferation and translation and whether the effect is reproducible in other, non-transformed cell lines.

Given these very supportive and constructive comments, we would like to invite you to revise your manuscript with the understanding that the referee concerns (as detailed above and in their reports) must be fully addressed and their suggestions taken on board. Please address all referee concerns in a complete point-by-point response. Acceptance of the manuscript will depend on a positive outcome of a second round of review. It is EMBO reports policy to allow a single round of revision only and acceptance or rejection of the manuscript will therefore depend on the completeness of your responses included in the next, final version of the manuscript.

We realize that it is difficult to revise to a specific deadline. In the interest of protecting the conceptual advance provided by the work, we recommend a revision within 3 months (August 8th). Please discuss the revision progress ahead of this time with the editor if you require more time to complete the revisions.

I am also happy to discuss the revision further via e-mail or a video call, if you wish.

*****IMPORTANT NOTE:

We perform an initial quality control of all revised manuscripts before re-review. Your manuscript will FAIL this control and the handling will be DELAYED if the following APPLIES:

- 1) A data availability section providing access to data deposited in public databases is missing. If you have not deposited any data, please add a sentence to the data availability section that explains that.
- 2) Your manuscript contains statistics and error bars based on $n=2$. Please use scatter blots in these cases. No statistics should be calculated if $n=2$.

When submitting your revised manuscript, please carefully review the instructions that follow below. Failure to include requested items will delay the evaluation of your revision.*****

- 1) a .docx formatted version of the manuscript text (including legends for main figures, EV figures and tables). Please make sure that the changes are highlighted to be clearly visible.
- 2) individual production quality figure files as .eps, .tif, .jpg (one file per figure). Please download our Figure Preparation Guidelines (figure preparation pdf) from our Author Guidelines pages <https://www.embopress.org/page/journal/14693178/authorguide> for more info on how to prepare your figures.
- 3) a .docx formatted letter INCLUDING the reviewers' reports and your detailed point-by-point responses to their comments. As part of the EMBO Press transparent editorial process, the point-by-point response is part of the Review Process File (RPF), which will be published alongside your paper.
- 4) a complete author checklist, which you can download from our author guidelines (<<https://www.embopress.org/page/journal/14693178/authorguide>>). Please insert information in the checklist that is also reflected in the manuscript. The completed author checklist will also be part of the RPF.
- 5) Please note that all corresponding authors are required to supply an ORCID ID for their name upon submission of a revised manuscript (<<https://orcid.org/>>). Please find instructions on how to link your ORCID ID to your account in our manuscript tracking system in our Author guidelines (<<https://www.embopress.org/page/journal/14693178/authorguide#authorshipguidelines>>)
- 6) We replaced Supplementary Information with Expanded View (EV) Figures and Tables that are collapsible/expandable online. A maximum of 5 EV Figures can be typeset. EV Figures should be cited as "Figure EV1, Figure EV2" etc... in the text and their respective legends should be included in the main text after the legends of regular figures.

7) Before submitting your revision, primary datasets (and computer code, where appropriate) produced in this study need to be deposited in an appropriate public database (see <<https://www.embopress.org/page/journal/14693178/authorguide#dataavailability>>).

Specifically, we would kindly ask you to provide public access to the scRNA-seq dataset.

The accession numbers and database should be listed in a formal "Data Availability " section (placed after Materials & Method) that follows the model below (see also <<https://www.embopress.org/page/journal/14693178/authorguide#dataavailability>>). Please note that the Data Availability Section is restricted to new primary data that are part of this study.

Data availability

Additional information on source data and instruction on how to label the files are available <<https://www.embopress.org/page/journal/14693178/authorguide#sourcedata>>.

10) Figure legends and data quantification:

- the name of the statistical test used to generate error bars and P values,
- the number (n) of independent experiments (please specify technical or biological replicates) underlying each data point,
- the nature of the bars and error bars (s.d., s.e.m.)
- If the data are obtained from n {less than or equal to} 5, show the individual data points in addition to the SD or SEM.
- If the data are obtained from n {less than or equal to} 2, use scatter blots showing the individual data points.

11) Our journal encourages inclusion of *data citations in the reference list* to directly cite datasets that were re-used and obtained from public databases. Data citations in the article text are distinct from normal bibliographical citations and should directly link to the database records from which the data can be accessed. In the main text, data citations are formatted as

follows: "Data ref: Smith et al, 2001" or "Data ref: NCBI Sequence Read Archive PRJNA342805, 2017". In the Reference list, data citations must be labeled with "[DATASET]". A data reference must provide the database name, accession number/identifiers and a resolvable link to the landing page from which the data can be accessed at the end of the reference. Further instructions are available at <<https://www.embopress.org/page/journal/14693178/authorguide#referencesformat>>.

12) As part of the EMBO publication's Transparent Editorial Process, EMBO reports publishes online a Review Process File to accompany accepted manuscripts. This File will be published in conjunction with your paper and will include the referee reports, your point-by-point response and all pertinent correspondence relating to the manuscript.

Kind regards,

Referee #1:

Review overview: Dr. Liu and colleagues investigated the relationship between CHOP expression and cell fate under endoplasmic reticulum (ER) stress using flow cytometry and single-cell RNA sequencing. They developed a novel conditional knock-out CHOP MEF cell line and tested the role of CHOP in proliferation of MEF cell in response to sub-lethal levels of ER stress-inducing agents, tunicamycin and thapsigargin. The experiment compared the incorporation of EdU in wild-type and CHOP knockout cells treated with sub-lethal ER stress inducers. The authors found that CHOP expression patterns differed in response to different levels of ER stress, with mild stress resulting in transient CHOP expression and severe stress leading to persistent expression. They identified two major groups of cells with distinct gene expression profiles from scRNA-seq analysis, suggesting that different CHOP expression patterns may lead to distinct cellular outcomes. The authors ascribe a role for CHOP in promoting proliferation/recovery in addition to its ability to promote cell death. The paper is well written and well-organized. The new flox-ed Chop allele may provide higher resolution insight into mechanisms of ER stress-induced damage. There are some concerns that should be revised, including a missing method protocol and overinterpretation of the results of the EdU incorporation experiment:

The major concern is that the experiments to study CHOP function at sub-lethal levels of ER stress all rely on non-specific ER toxins, tunicamycin and thapsigargin, that likely impact many other UPR pathways and cellular proteostatic mechanisms as well as directly damaging proteins and structures (glycosylation problems, ER and cellular calcium dysregulation). How do authors discern if the effects they are seeing (e.g., increased proliferation markers) are due to CHOP vs compensation by other UPR or proteostatic signaling molecules vs combination of multiple cellular/molecular changes caused by the toxins? To get a clearer idea, it would be helpful for authors to characterize how other UPR and proteostatic signaling molecules/pathways are impacted in their new CHOPfl/fl cells under sub-lethal ER stress conditions. It would be optimal to evaluate CHOP's functions, transcriptional targets, mechanisms independent of non-specific ER stress toxins

Other suggestions include:

1. In Figure 1, the results showed that EdU incorporation was higher in wild-type cells than in CHOP knockout cells during treatment with thapsigargin, indicating that CHOP supports EdU incorporation during ER stress. However, it should be noted that the lower EdU incorporation in CHOP KO cells could also be due to increased cell death. Therefore, it is suggested that the results be validated using other proliferation markers, such as Ki67 and gene expression markers tested in the single-cell sequencing analysis.

2. In the Discussion section, the authors should explore the possible reasons why CHOP appears to be necessary for DNA replication under ER stress only in MEFs and not in immortalized 3T3 fibroblasts. If similar results can be obtained in other cell types, they should be included in the discussion.
3. There are some missing protocols in method section for LDH cytotoxicity assay (Figure 2H), Adeno virus preparation (Figure 2F), liver immunostaining (Figure 6C) which should be included in the paper. In addition, please add CHOP expression vector information in Adeno virus.
4. Finally, thapsigargin treatment conditions (time and concentration) were different in Figures 1, 6 with in Figure 7 (ChopFLuL/FLuL, 10nM TG for 16 hours). Please address the reasons in the results section of the paper. If the same condition might cause different results, please discuss the reason in discussion section.
5. There are many reports that CHOP^{-/-} animals do not rescue phenotypes and pathologies when bred in various animal disease models linked to ER stress. These should be included and discussed as they support the notion that CHOP is not exclusively involved in or important for cell death.

Referee #2:

In this work, Liu, Zhao, and Rutkowski show opposing roles for the long-recognized proapoptotic transcription factor CHOP in driving an adaptive phenotype. The authors developed an elegant gene trap-based model system to dissect these roles based on targeted CHOP expression and deletion in the same cells, which provides a robust experimental system. The authors also provide compelling evidence for CHOP roles in cell cycle progression. Finally, the authors propose a model wherein CHOP functions as a "stress test" in cells enduring physiological stress, which preserves cells able to restore homeostasis and eliminate those that cannot. While an exciting finding, some aspects of the model are difficult to accept due to insufficient experimental evidence or data interpretation, specifically regarding stimulation of protein synthesis by CHOP and CHOP-dependent and -independent transcriptional programs obtained by scRNASeq. While the findings are appropriate for the readership of EMBO Reports, the manuscript is not ready for publication in its current form, and the following issues need to be addressed.

Major issues

1. The most convincing data shows a CHOP effect on cell proliferation. However, additional experiments are needed to yield mechanistic insights. For example, cell cycle synchronization coupled with analyses of cyclin levels (mRNA and protein) can reveal the extent of dilation of the cell cycle in the absence of CHOP and which stages are primarily affected. In addition, the authors should discuss why the effect of CHOP is lost in transformed cells (NIH 3T3). Is the CHOP cell cycle effect preserved in human cells? Is it also lost in transformed human cells?
2. The authors claim that CHOP supports proliferation by restoration of protein synthesis. This conclusion is drawn from published work in different model systems and the use of the ISR inhibitor ISRIB in cell culture. Experiments showing protein synthesis restoration should be carried out in this experimental system (metabolic labeling, puromycylation of nascent peptides, BONCAT, etc.), and orthogonal polysome profiles could be used to validate the conclusions. Because CHOP induces GADD34, it is desirable to directly assess its role in restoring protein synthesis by blocking GADD34 activity using siRNAs. The effects stemming from the action of CREP can be ascertained in the same way. Moreover, pharmacological agents can be deployed in orthogonal experiments to support their claims (e.g., sephin, guanabenz, salubrinal).
3. While the scRNASeq data provides interesting insights, it is difficult to accept the phenotypic groupings based on the analysis of hand-picked genes. A few examples illustrate this point. Rather than looking at individual chaperones, it would have been more informative to query gene expression signatures for ATF4, ATF6, and XBP1 target genes, which are available in the literature. In the discussion, the authors claim that cells with an activated UPR were confined to groups 1, 2, 3, 5, 9, and 13. However, the analyses in Fig. S3 suggest UPR activation in groups 2, 9, and 13, while groups 1, 3, and 5 seem enriched in extracellular matrix organization and cell migration processes. The authors also argue that CHOP exacerbates ER stress and prolongs UPR activity (IRE1 activation). However, an alternative explanation is that a gene expression program downstream of CHOP is in place to alleviate ER stress by increasing UPR signaling and enhancing XBP1 splicing (as shown in Fig. 6D), which is mainly a pro-survival outcome. These lines of argumentation need to be reconciled. Last, the observation that ribosomal biogenesis is upregulated in cells devoid of CHOP warrants further discussion, specifically considering the claimed role of CHOP in regulating protein biosynthesis.

Minor issues

1. The result with TM in Fig. 1C shows a modest effect. Please consider showing different drug concentrations and provide FACS scatter plot data to show the effect on the cell population.
2. Consider revising the schematic of the gene trap to indicate the locations of the CHOP uORFs, and clarify their roles in the

narrative.

3. In Fig. 2F, please explain the difference in mRNA levels when measuring exons 1-2. Is this effect due to different mRNA half-lives?
4. Please show unstressed control in Fig. 2G.
5. Consider using consistent drug doses throughout the manuscript.
6. Please acknowledge a modest increase in basal EdU incorporation in non-stressed CHOPFLuL/FLuL cells in panel 3A compared to w.t. (20.1% vs. 22.4%). Consider showing FACS scatter plots for data in panels C and D to show the effect on the cell population.
7. In panel 4B, the authors claim no difference between w.t. untreated and Q-VD treated cells, even though the bar graphs show a modest effect.
8. 5 mM DTT is a very high drug concentration that could lead to pleiotropic effects. Please provide a rationale for its use at this concentration.
9. Consider showing histograms for DNA content in panel 4C.
10. Please show blots for P-eIF2a, ATF4, and GADD34 in panel 4E.
11. The effect of ISRIB shown in Fig. S2 is too modest to conclude that "protein synthesis was possibly more strongly inhibited by TG in CHOP-null cells."
12. The claim that "ISRIB increased EdU incorporation upon TG treatment in CHOP-null cells to levels comparable to wild-type" is inaccurate, as illustrated in the bar graph (rel. incorporation is >50% in w.t. while in cells devoid of CHOP is <50%).
13. Please quantify the percentage of XBP1 splicing in Fig. 6D and correct the figure pointer on the second page of the discussion; there is no Fig. 6E.
14. Please include the gene tables at the end of the results section "Applying this logic, we tabulated the genes...."
15. Show expression levels on the UMAPs in Fig. 7A in the same scale (nLuc v. Chop), and clarify the groupings in the first UMAP by increasing font sizes. Include clusters exemplifying additional genes, for example, those associated with ATF4, ATF6, and XBP1 gene expression signatures.

Referee #3:

The authors describe a novel role for the stress associated transcription factor Chop in promoting cellular proliferation during endoplasmic reticulum (ER) stress conditions using pharmacologic inducers of ER stress in combination with genetic manipulations of Chop expression. Herein, they describe a new set of expression constructs that permit close control over the expression of the stress associated transcription factor Chop. Using single cell sequencing, they identify stress associated gene targets that are correlative and/or anti-correlative with Chop expression. This work suggests that CHOP functions as a 'stress test' to dictate adaptation or apoptosis following varying levels of ER stress.

Overall, this is an interesting and important manuscript that puts into balance the long-held view of CHOP as 'pro-apoptotic' transcription factor. The experiments are well designed and executed and generally support the overall conclusions of the manuscript. That being said, there are a few places where I think the manuscript could be improved. A bit more focus on the mechanistic basis of the observed CHOP-dependent increases in proliferation would be useful, specifically as it relates to the importance of CHOP-dependent regulation of protein translation. The ISRIB experiments are nice, but it would be interesting to show that similar phenotypes are observed in other available cells where resolution of stress-induced translation attenuation is impaired (e.g., GADD34-deficient MEFs). That would help further support the conclusion that it's the restoration of translation downstream of CHOP that is important for proliferation. Further, it is important to test the CHOP-dependent proliferation experiments in additional cell lines, especially as this was observed in MEFs, but not 3T3 cells. I realize that this could easily be something related to the immortalization of the 3T3 line, but, considering that result, showing other cell lines (preferably derived from other tissues) demonstrate similar effects would strengthen the findings and potential significance of this work. It could also be interesting to study whether similar competitive advantages are observed under stress conditions that activate other ISR kinases (apart from PERK), which could also expand the scope of this finding. However, this could be pursued in subsequent studies. Ultimately, this is a solid manuscript that provides a much needed response to the long-held (I think somewhat misguided) view of CHOP as a pro-apoptotic transcription factor, instead painting the role of this protein as a 'stress test' during variable levels of ER stress. I would support publication of a revised manuscript that addresses the above points, as well as some of the additional points described below.

Overall strengths:

- The FLuL, floxed, and KO allele approaches for modulating Chop expression provide a sensitive & specific approach for regulating Chop expression.
- The use of EdU in combination with PARP cleavage and EdU + propidium iodide provide clear evidence of the distinct proliferative and apoptotic cell populations associated with Chop expression.
- The proliferation assay in Figure 5 provides insights into the competitive advantages of cells capable of proliferating during conditions of modest stress.
- The use of three distinct ER stressors (Tg, Tm, and DTT) is laudable in showing the extent to which CHOP impacts proliferation in the context of a variety of ER stress conditions.

Overall weaknesses:

- The proliferative effects shown in Fig. 1A, Fig. 3A, and 4C are relatively weak overall and strongly influenced by the total events collected for each sample, which will impact the overall proportion of cells appearing in each population. (Consistent with this, the FSC-A gating in Figs. 1A and 3C appears to cut off a portion of the main cell population. The validity of this gating approach must be addressed [in the supplement] to substantiate the conclusions drawn from these key figures.)
- While the use of propidium in combination with EdU is laudable, there appears to be fewer cells total in the ChopFluL/FluL condition. Quantification across multiple experiments normalized to cells collected per experiment (with relevant statistical testing). Given that a major finding of the paper is that Chop depletion reduces proliferative as well as apoptotic cell populations, this key experiment should have additional replicates or other supporting evidence (e.g. supplemental Tg, Tm experiments).

Minor critiques:

- The statistical testing used for the majority of calculations may not be appropriate. Based on the calculated significance in several figures despite substantial overlap, normality and/or equal standard deviation should not be assumed between conditions. Prism automatically performs SD equivalency testing when running ANOVAs. Please either correct statistical testing, or otherwise ensure that the Brown-Forsythe/Bartlett p-values are greater than 0.05, esp. for Fig. 1B- 1D, 4F, and 7D.
- The extent of Chop induction by treatment with 5mM DTT for 2h is not shown (Fig. 4C), whereas the protein levels for Tg and Tm are depicted within the manuscript.
- Consider rewriting Fig. 7 section of text to better emphasize the relationship between overall enrichment of ER stress-linked genes and specific expression of CHOP in these clusters.
- Improving the clarity of the writing regarding the subtle distinctions between ER stress, UPR activation, and chaperone expression would benefit the readability of the paper and better highlight the important interesting contributions this study brings to the field.
- The color scale of the bars for Fig. S3 is dependent on the total number of genes in the gene set and should be scaled as such.
- o It would be more informative to provide the top DEGs that define each cluster, rather than the relative GO enrichment for each cluster.
- Please include statistics (e.g. mean, sd) for number of reads per cell for each sample included in single cell sequencing dataset.

Minor corrections:

- The specific loading control protein should be listed for Fig. 2E, 2G, 4E.
- Axis labels are missing from Fig. 4D
- Fig. 7C: reconsider the use of black as the maximal value on a dark gray background.
- The references for Figs. 2H, 4D, 7F are not included in the appropriate sections of text. Fig. S2A is not included in main text but is highlighted as a major finding in the discussion.
- Missing a quote at the end of "DNA-dependent DNA replication" in text
- The corresponding author has two institutional notations (1,3), but the second one is not listed.

Liu et al., Response to Reviewers

We thank the reviewers and editor for their comments on our manuscript. We are grateful that they all appreciated the novelty and significance of the work, and we have extensively revised the manuscript to address their comments, including through the addition of a considerable body of additional data. The most significant point, raised in one form or another by all three reviewers and by the editor, was the question of whether the CHOP-dependent proliferation phenotype that we have identified was generalizable to other cell types beyond MEFs, particularly in light of the data from CHOP-deleted 3T3 cells present in the original manuscript. We have addressed this concern by showing, using multiple separate control or CHOP-deleted clones of a liver cell line (TIB-73), that the proliferation function is observed in these cells as well. These data are now presented in Figure 1D and EV1F. Moreover, we also found that, even after immortalizing our primary

MEF cell lines, the CHOP-dependent proliferation phenotype was maintained (although we do not include that result in the present manuscript, as the TIB-73 result shows that both a non-fibroblast and immortalized cell type preserves the phenotype of interest). Therefore, it appears that, while it is still possible that some cell types such as 3T3s do not reflect this function of CHOP (for as yet unknown reasons that lie beyond the scope of this paper), nor is the result restricted to primary MEFs.

Further responses are provided below.

Reviewer 1

The reviewer raises the concern that the proliferative function of CHOP was elicited using non-specific ER toxins such as tunicamycin and thapsigargin, that will affect ER proteostasis and UPR signaling beyond just CHOP, and they suggest both that we better characterize how the UPR is impacted by these toxins and that we evaluate CHOP's function independent of such toxins.

We thank the reviewer for raising this point, as we are well-attuned to the non-physiologic nature of drugs like TM, TG, and DTT that, useful though they are, are still imperfect representations of physiologic stimuli. The central issue, though, is that, as best we or the literature in the field can determine, CHOP's functions are largely or wholly dependent on some sort of discrete stress stimulus. In the absence of such as stimulus, CHOP protein is undetectable (Fig. 2E, EV1G) and its RNA expression very, very low (typical Ct values are ~30). In other work, we have also performed RNA-seq on livers of wild-type or CHOP-deleted animals and, absent an ER stress stimulus, there are almost no significant CHOP-dependent changes in gene expression (data not shown here but can be provided for inspection if necessary). This stress-dependence makes us reliant on tools like TM, TG, and DTT to probe CHOP's function. We did attempt to use virally-mediated overexpression of a constitutively misfolded ER client protein (we showed the ability of this substrate, NHK, to elicit ER stress in our previous paper, Gansemer et al., 2020, *iScience*), but the magnitude of ER stress induced by this virus was sufficiently small that it was impossible to test a CHOP-dependence of proliferation. While we agree that pharmacological toxins are suboptimal tools, the fact that the CHOP-dependent proliferation phenotype is observed for TG, TM, and DTT—which elicit ER stress by three distinct mechanisms—and at various treatment regimens, means at least that the result is very likely to reflect a true ER stress-dependent function of CHOP rather than an artifactual effect of any one drug.

In general, we agree that agents like TG, TM, etc. have wide impacts on ER proteostasis. The general effects of these drugs on cultured cells in general and primary MEFs in particular have been well-documented over an extensive body of literature over the last 20-30 years. The more focused question of how these responses are altered when CHOP is absent has likewise been the subject of extensive study, from which it is apparent that CHOP potentiates cell death during severe ER stress. And excellent work from the Ron, Kaufman, and Hatzoglou labs in particular has shown that CHOP's effects on cell death are best rationalized by its effects on protein synthesis. Where our manuscript sets a very different course from that prior body of work is in examining an outcome (proliferation) that has not been previously examined, in combination with stress conditions (mild ER stresses, rather than overwhelmingly cytotoxic ones) where that outcome is salient. Our data give us no reason to believe that the effects of CHOP on proliferation under these circumstances are due to any previously unheard-of molecular function of CHOP, but rather that these effects—like those of CHOP on cell death—emerge from CHOP's promotion of protein synthesis; this conclusion is supported both by our original ISRIB data and now our added Salubrinal and *Gadd34* KD data (see below). However, one of the claims of this manuscript, very much in line with the comment of the reviewer about the UPR and other proteostatic signaling molecules, is that the promotion of protein synthesis by CHOP has in turn effects on the persistence of UPR signaling. This point was originally supported by data from Figures 6 and 7, and we have added new data, most notably including a wider analysis of UPR target gene expression from scRNA-seq data (Fig. 7E and EV4F) that better define this relationship.

Reviewer 1 Other Issues

The reviewer raises the concern that “the lower EdU incorporation in CHOP KO cells could also be due to increased cell death” and suggests that other proliferation markers be examined.

Although it could be possible in principle that increased cell death could explain lower EdU incorporation, CHOP KO cells exhibit less cell death during ER stress, not more (as shown in Fig. 2G, 2H, and in many prior published papers that point to a role for CHOP in cell death). However, we have also added cell cycle analysis (Fig. EV1D) and examination of various cell-cycle-related genes (Fig. EV1E). Although these results do not point definitively to a particular point of control over the cell cycle exerted by CHOP—which is perhaps unsurprising since CHOP is not a classical cell cycle regulator—they do support that there are significant alterations to the machinery of cell cycle progression, such as suppression of *Pcna*, *Cyclin A2*, and *Cyclin E1*, and dilation of the G2 phase of cell cycle, that occur as a consequence of CHOP action.

The reviewer suggests an account in the discussion section of the discrepancy between primary fibroblasts and 3T3s as well as results from other cell types.

As we have removed the 3T3 data (see above), a discussion of the lack of the phenotype in those cells is now moot. As detailed above, we have found the proliferation phenotype to hold also in TIB-73 cells, and those data are included in both Result section, and a more general treatment of the role of proliferative capacity in a potential in vivo role for CHOP is provided in the Discussion.

Reviewer requests added Methods Details for LDH assay, Adeno preparation, and liver immunostaining.

We have modified the Methods to either add these details (for LDH and Adeno preparation) or to make their inclusion more obvious (liver immunostaining was previously listed as having been done in the same manner as a previous paper of ours; that reference remains but the section has been retitled to make its inclusion clearer).

Reviewer requests that we add CHOP expression vector information in Adenovirus.

We are uncertain what is meant by this request, because the manuscript does not use any CHOP expression vectors. CHOP was manipulated in MEFs solely by treatment with either adenovirus expressing FLPo, which deletes the gene trap to permit CHOP expression, or expressing CRE to delete the CHOP ORF and thereby prevent CHOP expression. These methods are presented in detail in the Methods section.

Reviewer requests that reasons for choosing different concentrations of TG be included.

We have amended the text describing Figures 6 and 7 to state explicitly why we selected the doses that we did. We also note that the EdU phenotype is remarkably robust across a range of stress conditions (different stressors, different doses).

Reviewer recommends discussion of situations in which knockout of CHOP fails to rescue phenotypes and pathologies when bred in various animal disease models linked to ER stress.

We agree with the reviewer that the observation that deletion of CHOP fails to rescue some ER stress-related phenotypes argues for a more complex role for this protein than is appreciated. However, failure of CHOP deletion to protect against any given pathology could be for any number of reasons and is therefore difficult to interpret. Even if CHOP had no other role than killing cells (a proposition we obviously do not support), its deletion might fail to rescue an ER stress-related pathology simply if there were a redundant stress-induced death pathway. We have added text to the introduction noting that deletion of CHOP protects against many—but not all—ER stress pathologies, but feel that a lengthier discussion of instances where CHOP deletion does not protect might leave the perception that we are being too speculative about CHOP's in vivo roles beyond what the present manuscript can justify.

Reviewer 2

The reviewer requests additional experiments to examine cell cycle progression in wild-type versus CHOP-null cells.

We have clarified in the manuscript that the original data from Fig. 1B was obtained using synchronized cells. We have also added TG data from non-synchronized cells (new Fig. EV1B) and also mention that TM results from synchronized cells were also the same—in short, the phenomenon is extremely robust. We also provide qRT-PCR analysis of cell cycle genes and FACS analysis of cell cycle stage. We note, though, that CHOP is not itself likely to be a cell cycle regulator per se, but rather to permit cells to progress through the cell cycle due to their resolution of stress. For this reason, while the additional cell cycle data demonstrate that, indeed, the presence or absence of CHOP affects cell cycle pathways, we wish to be cautious not to over-interpret these new findings.

Reviewer requests that we discuss why the CHOP effect is seemingly lost in 3T3 cells and test other cell types

As mentioned above, we found that TIB-73 liver cells (which are immortal) maintain the CHOP effect (Fig. 1D), demonstrating that it is not restricted to just fibroblasts, nor to just non-immortalized cells. Thus, while some cells lines, or at least apparently 3T3s, do not maintain the CHOP effect, this is neither a consequence of being non-fibroblasts nor of being immortalized (and, also as mentioned above, we found our MEFs maintained the CHOP effect even after immortalization, although we chose not to include these data). We note that we used TIB-73, which are mouse cells, simply because we already had validated gRNAs for targeting mouse CHOP by CRISPR, but that we have no reason to believe that there are not human cells that would behave similarly.

Reviewer requests orthogonal examination of the hypothesis that CHOP promotes proliferation through protein synthesis

In addition to the ISRIB data mentioned by the reviewer, the original manuscript also contained metabolic labeling, in original Figure S2 (now EV3F, G), which was an approach that the reviewer suggested. These data show very clearly that ISRIB, as expected, increases protein synthesis in TG-treated cells. We also tested the effects of GADD34 knockdown and Salubrinal, as suggested, and those experiments further validate the idea that proliferation arises as a consequence of protein synthesis, because both GADD34 knockdown and Salubrinal (which inhibits GADD34) diminish EdU incorporation (new Figs. 4E-G). It is worth noting that, due to the complex feedbacks and redundancies within the eIF2 α signaling axis, CHOP is not the only component that controls protein synthesis, and so differential effects of any of these manipulations on wild-type versus CHOP-null cells are difficult to predict. For example, one might predict that salubrinal would have no effects in CHOP-null cells because CHOP regulates GADD34, and so, to that line of thinking, there should be no GADD34 to inhibit in CHOP-null cells. However, GADD34 is also transcriptionally regulated by ATF4 and translationally regulated by eIF2 α phosphorylation. For this reason, we interpret our results in the simplest manner, that ISRIB promotes protein synthesis and also promotes proliferation at least in CHOP-null cells, in which protein synthesis is otherwise lowest during stress. Conversely, Salubrinal is known to prolong the repression of protein synthesis and also inhibits proliferation. Thus, there is a simple relationship between promotion of protein synthesis and proliferation that is most evident in the cells (CHOP-null) in which one of the regulators of this relationship has been severed. But the ambiguities inherent in this line of experimentation are the reason that we state that “the proliferative function of CHOP is carried out at least in part by its previously described role in reversing the attenuation of protein synthesis caused by eIF2 α phosphorylation.”

As recommended, we also tried Sephin and Guanabenz but found that these drugs are acutely toxic to primary MEFs at least in our hands. We are unsure of the reason for this (much of the original characterization of these two drugs was performed in HeLa and other cell lines, though some experiments were also conducted in MEFs; whether these were primary or immortalized MEFs was not clear) but it is an observation we have made before. Nonetheless, we feel that the added Sal and GADD34 KD data support our conclusions.

Reviewer requests a more extensive analysis, reconfiguring, and reinterpretation of scRNA-seq data

We have extensively reworked the description of the scRNA-seq data in both the Results and Discussions sections to take into account this critique, and have replaced and/or added new data to strengthen the support for our conclusions. Specifically:

- Figure 7E and EV4F now contain a more extensive analysis of UPR target genes, selected based on the diversity in their mechanisms of regulation (i.e., genes that have been previously attributed to IRE1, ATF6, or PERK signaling or various combinations thereof—more on this below in this response). We show in new Figure EV4F how these genes are expressed in each of the 15 clusters, and in modified

Figure 7E the extent to which the expression of each these correlates with CHOP vs. nLuc across the total data set. The data—particularly the correlation data—show that the large majority of these genes correlate with CHOP more strongly than with nLuc. Given that both CHOP and nLuc are positively regulated by the UPR, their stronger correlation with CHOP suggests that a component of their upregulation is attributable to CHOP specifically more than UPR activation generally. Notably, this holds true both for genes that are likely direct transcriptional targets of CHOP (*Gadd34*, *Trib3*, etc.) and genes that are likely not. (We base these attributions on the very thorough global analysis of CHOP transcriptional targets in MEFs previously conducted by the Kaufman lab). Therefore, the data are consistent with the idea that CHOP activity leads to global persistence of UPR activation

- Related to the point above, the reviewer points out that this persistent activation could be a consequence of CHOP exacerbating ER stress, as we propose, but also could be a consequence of CHOP acting on UPR signaling independent of ER stress. We agree with the reviewer that our data do not definitively distinguish between these two possibilities, and we more explicitly account for them in the modified Discussion section to account for why we favor the former possibility and to acknowledge that, absent direct, very sensitive single-cell-specific assays for ER proteostasis, the question must for now remain open.
- We have also streamlined the Results and Discussion to hopefully clarify what we mean when we talk of the clusters that are diagnostic of UPR activation—that principally this is based on the pathway analysis shown in Figure 7B that indicates clusters 1, 2, 3, 5, 9, 10, 13, and 14 as enriched in “response to ER stress” genes. The expression data in Figure EV4F support this assignment, as all but 3 of the 20 UPR target genes examined are maximally expressed in one of these groups. Although our data show, as the reviewer points out, that clusters 3 and 5 are more associated with cytoskeletal genes than ER stress ones (Figure 7B), our intent was to include all of the clusters that show genome-wide evidence of UPR activation, rather than to identify only those in which UPR activation is the single most dominant pathway (indeed, as Figure EV4E shows, only clusters 2 and 9 would fit this more stringent criterion).
- We have also added a more explicit treatment in the Discussion about the seemingly paradoxical observation, as pointed out by the reviewer, that CHOP is associated with stimulation of protein synthesis but also apparent suppression (at least in terms of regulated genes) of ribosome biogenesis. While we do not yet know the import of this finding, it is consistent with our previous work (DeZwaan-McCabe et al., 2013, PLoS Genetics), and we point this out and speculate on it.

While we have expanded our analysis of UPR target genes to more expansively include those regulated by different UPR pathways or combinations of pathways, we feel from our own experience and a broad view of the literature that grouping genes as ATF6-, IRE1-, or PERK-dependent probably greatly oversimplifies gene regulation within the UPR. Even prior to the data in this manuscript, it was clear that the various UPR pathways overlap in their outputs in ways that defy simple categorization. For example, classical IRE1-dependent targets containing ERSEs (*Edem1* being the most oft-cited example) are also regulated by ATF6; likewise, PERK appears to greatly impact the expression of the vast majority of UPR targets, even those that are not thought to lie directly downstream of it. The present work highlights this complexity as well, as the ER chaperones BiP and Grp94 have widely been considered to share mechanisms of regulation, and yet their expression is clearly separable in our scRNA-seq analysis (Fig. 7D). We now comment explicitly on this point in the Discussion (“Likewise, the broader analysis of UPR target genes in Fig. EV4F shows...”)

Together, we hope that these changes provide greater clarity and better illustrate why we come to conclusions that we do.

Reviewer 2 Other Issues

The reviewer requests differing drug concentrations and scatter plots for FACS data in Figure 1

Although the reviewer is correct that the effect of CHOP deletion on proliferation in the TM treatment regimen is modest, this is largely because the stress exerts such a strong net effect on proliferation. It is worth noting that, although only a minor population of wild-type cells is EdU positive under those conditions, almost no *Chop*^{-/-} cells are (FACS plots added in Fig. EV1C). We also note that the phenomenon is extremely robust, occurring in response to different doses of TG (10, 25, and 50 nM TG are shown in Fig. 3C, D, EV4A) and different stressors (TG through most of the paper, TM for Fig. 1C/EV1C, DTT for Fig. 4C and EV3A-C). We also show that the difference is seen in both synchronized cells (Fig. 1B) and non-synchronized cells (Fig. EV1B).

Reviewer asks for revision of gene trap schematic

Done as requested

Reviewer asks to account for why Exon 1-2 expression is higher in FLPo-treated cells than control (i.e., knockout) cells

Results describing Fig. 2 modified as requested

Reviewer requests unstressed control be shown for Fig. 2G

Done as requested; Fig. EV1G

Reviewer requests use of consistent drug dosing

We agree that this would be ideal, but would essentially entail repeating all of the TG experiments at 10 mM (the dose used for the scRNA-seq experiment, for which we now provide additional data (Fig. 6A) showing the same CHOP-dependent proliferation phenotype). Given the robustness of the phenotype to different stressors and doses, as described above, we feel that using different doses is not intrinsically problematic and is even a strength of the presentation. Moreover, we have modified the text to be clearer, particularly for Figures 6 and 7, about why we chose the doses that we did for those experiments.

Reviewer requests acknowledgment of an increase in basal EdU incorporation in stressed FLuL/FLuL cells in Fig. 3A, and to show FACS plots for 3C and D

The data shown in Fig. 3A are representative images and the variance observed is within the normal range of variation of the experiment; the absolute amount of basal EdU incorporation varies based on the line chosen, the day, etc. We have not observed any systematic genotype-specific differences in basal EdU incorporation. We now show representative FACS plots for 3C and 3D in Fig. EV2.

Reviewer notes that Q-VD has a modest effect on EdU incorporation, independent of genotype

This is correct, and we have modified the figure legend to note that we were not showing all possible pairwise statistical combinations, but rather just comparisons between genotypes, with the point being that, if the EdU signal were attributable to dying cells (directly or indirectly), then inhibiting cell death with Q-VD would eliminate the genotype-specific difference in EdU incorporation—which it does not do.

Reviewer requests a rationale for use of 5 mM DTT

5 mM is a high dose of DTT but, unlike TM or TG, DTT is pharmacologically reversible and can be washed out. High doses are needed to elicit ER stress over such a short time frame (2 hours of treatment). As we now note in the Results, we have previously shown (Wu, Rutkowski, et al., 2007, Dev Cell) that MEFs can adapt to and proliferate despite repeated treatment with DTT (in that case, repeated 1h treatments of 10 mM). We also now show in Fig. EV3D how CHOP expression responds to different DTT pulsing paradigms.

Reviewer requests DNA histograms for panel 4C

Done as requested; Fig. EV3A

Reviewer requests blots for PeIF2 α , ATF4, and GADD34 in panel 4E

We provide a new blot showing PeIF2 α and ATF4 (in addition to CHOP). Unfortunately, we could not identify an antibody that provided a validated specific signal to detect GADD34 in MEFs; the antibody we used to use for that purpose is no longer made by the manufacturer. However, the other additional blots provide a fuller picture of the effects of ISRIB on the ISR.

Reviewer objects to two statements concerning ISRIB effects on cells

The statements have been deleted or modified

Reviewer requests quantification of Xbp1 splicing in Fig. 6D and notes lack of a Fig. 6E

We have provided quantification and statistical analysis in new Fig. 6E. We also note that we replaced original Fig. 6C with new panels, which differ in substance from the originals only in being counterstained with hematoxylin, so that we could also provide the source data.

Reviewer requests data tables underlying Fig. 7F

This data is provided in new Table EV2

Reviewer requests changes to Fig. 7A (font sizes of clusters and scales) and also requests that we tie individual clusters to UPR gene expression signatures

The changes to Fig. 7A were made as requested. And, as described above, we have conducted a more extensive analysis of UPR target gene behavior and how it is tied to the various clusters, particularly in new Fig. EV4F.

Reviewer 3

The reviewer requests that we augment ISRIB experiments with others showing similar phenotypes in, for example, GADD34-deficient cells

As described above, we have added data in GADD34-knockdown cells and using Salubrinal that support our conclusions about the role of translation

Reviewer states that it would be important to test CHOP-dependent proliferations effects in other cell lines

As described above, the revised manuscript validates the phenotype in TIB-73 liver cells, and we also validated that the phenotype is maintained upon immortalization in our MEF lines, though the latter data are not included.

Reviewer suggests that it could be informative to study stresses that activate other eIF2 α kinases besides PERK

We agree with the reviewer, and also with their comment that this could be pursued in subsequent work.

Reviewer requests that we justify our gating approach for FACS experiments

For this purpose, we use new Figures EV1A and EV3C to exemplify our gating approach, showing not only representative EdU FACS plots but also the gating selections that were applied so that the process is transparent. By adjusting the gating parameters for Figure 1A, we now include new data that avoids the “cut-off” appearance of the original Figure 1A, though the conclusions are unchanged. All the proper voltages were set up with the assistance of the staff from our flow core, using either a sample from the experiment or samples from a pilot experiment. We first gated on the main cell population to eliminate debris, and a second gating on the FSC-W channel was done to eliminate the doublets, so that all the EdU positive cells indicate a single proliferating cell. EdU, Cleaved-PARP, and CHOP gating were all determined by using proper non-primary controls (secondary only). Since the separation in EdU staining was quite obvious, not every single experiment had a EdU negative control. We avoided setting the EdU gate too close to the main population, because that would risk counting cells that were undergoing DNA repair. We have found empirically that minor alterations of the gating selections did not materially affect the results obtained. Further details on gating are now provided in the Methods section.

Reviewer requests expansion of the EdU/PI data and raises a concern that the finding could be an artifact of a different number of cells analyzed

New Figure EV3C is also included to address this concern—there we show that the extent of EdU positivity remains consistent even when the number of events analyzed is very different (in that case, ~10,000 vs. ~20,000). In our experience the differences in EdU positivity are robust to differences in cell count, which can be seen in part by the fact that the phenotype is maintained in experiments with large cell counts such as Fig. 4C and experiments with smaller cell counts such as Fig. 1C. We also now include Fig. EV3B to show an independent replication of the experiment from Fig. 4C. We did not feel it essential to replicate the experiment enough further times to carry out statistical analysis primarily because the major point of the panel is simply that the EdU-positive cells are in fact in the process of progressing through the cell cycle, which is qualitatively evident from the temporal progression of the data. This was to test (and eliminate) the possibility that EdU positive cells were replicating their DNA but not proliferating, which would have undercut our assertion that CHOP was providing a proliferative advantage. It is also qualitatively evident from this experiment that, as with TG and TM, *Chop*-null cells undergo less proliferation during DTT treatment, which is seen in both experiments at all three time points analyzed. We believe addition of this additional data set increases the rigor of the experiment.

Reviewer raises a concern about the normality of data in statistical comparisons

We thank the reviewer for raising this point. The reviewer is correct for the data in Figure 1B, and so the current manuscript uses a Mann-Whitney test instead, which, while it eliminates the statistical significance for the 2.5 nM TG condition, does not materially change the conclusion. Figures 1C and 4E (old 4F) satisfy the normality condition, and Figure 1D is new data that also satisfies the normality condition. New Figure 4G did not satisfy the normality condition, so for that we used a Welch's ANOVA, which is noted in the Methods. Figure 7D is not an ANOVA, but rather a Wilcoxon Test. We have edited the section in the Methods on Statistical Analysis, as well as the Figure Legends where applicable, to reflect this.

Reviewer requests that the effect of DTT on CHOP induction be shown

This is now provided from a similar experiment (the experiment we used to determine the dosing paradigm for DTT) in new Fig. EV3D.

Reviewer requests rewriting text describing Fig. 7 to more clearly link UPR target gene expression to CHOP

We have revised accordingly

Reviewer requests that the distinctions between ER stress, UPR activation, and ER proteostasis be more clearly drawn.

We agree that there can be a tendency to conflate these ideas (and the ISR as well) even though they are separate. We have modified the writing in the Introduction, Results, and Discussion, to hopefully make these ideas sharper. In fact, the edited Discussion includes an explicit treatment of the fact that our model, which proposes that CHOP causes persistent UPR activation by exacerbating ER stress, is materially different from a hypothesis in which CHOP directly regulates the UPR independent of ER proteostasis. We hope that the clarity has been improved.

Reviewer requests that the bars in Fig. EV4E (old Fig. S3) be color coded by percentage of gene set rather than raw gene count

Upon reflection, we feel that neither metric is not terribly meaningful in its own right, and that the single most important metric we wanted to convey was statistical significance. For that reason, the coloration differences in that figure have been eliminated.

Reviewer requests scRNA-seq quality control data

This is now provided in Fig. EV4B-D.

Reviewer 3 Minor Corrections

The reviewer raises several minor labeling/formatting issues

These have all been corrected.

Dear Tom,

Thank you for the submission of your revised manuscript to EMBO reports. We have now received the full set of referee reports that is copied below.

As you will see, all referees are very positive about the study and request only minor changes to clarify text and figures. I fully agree with referee 1 that the negative data on CHOP's role in NIH3T3 cells should be reported and remain part of the manuscript.

From the editorial side, there are also a few things that we need before we can proceed with the official acceptance of your study.

- Please provide up to 5 keywords.
 - Table EV1 and EV2 should be Dataset EV1 and EV2, respectively. Please provide a legend in a separate tab.
 - Please add the header 'Disclosure and competing interests statement' to the conflict of interest statement. For more information see <https://www.embopress.org/page/journal/14693178/authorguide#conflictsofinterest>
 - Please remove the Author Contributions from the manuscript file and make sure that the author contributions in our online submission system are correct and up-to-date. The information you specified in the system will be automatically retrieved and typeset into the article. You can enter additional information in the free text box provided, if you wish.
 - To comply with our editorial policies, please revisit the statement 'data not shown' on page 7. Please either show the relevant data or remove any statements that are based on data not included in the manuscript.
 - Funding information: please ensure that the information provided in the online submission system is congruent with that in the manuscript text. The information on "funds from the University of Iowa Department of Anatomy and Cell Biology" is missing in the submission system.
 - Data availability: please add a link that resolves directly to the deposited dataset.
 - "Experimental procedures" should be renamed to "Materials and Methods".
 - References should be placed before the figure legends.
 - Please do not calculate statistical significance based on two replicates (e.g., KO condition in Fig. 1D).
 - Please write the abstract in present tense.
 - Our production/data editors have asked you to clarify several points in the figure legends (see below). Please incorporate these changes in the manuscript and return the revised file with tracked changes with your final manuscript submission.
- a) Legends of Figure 1 and EV1: If information on statistics applies to more than one panel, please move/add the description to a separate 'Data Information' section at the end of the respective figure legend and indicate to which panels the information refers to.
- b) Please note that the expanded view figure legend style does not comply with the journal guidelines i.e all the EV figure legends are in a run-on style. Instead, each panel description should be separated by a line break.
- c) Please note that in figures 1b-d there is a mismatch between the annotated p values in the figure legend and the annotated p values in the figure file that should be corrected.
- d) Please define the annotated p values ****/***/**/* in the legends of figures 2f, h; 3b-d; 4b, e-g; 5d-e; 6e; EV1b, d, e; EV3e; EV4a as appropriate.
- e) Please indicate the statistical test used for data analysis in the legends of figures 7b, f; EV4e.
- f) Please note that information related to n is missing in the legend of figure 2f, h; 3b-d; 4b, e-g; 5b-e; 6e; 7d; EV1b, d, e; EV3e, g; EV4a-d

g) Please note that the error bars are not defined in the legend of figures 2f, h; 3b-d; 4b, e-g; 5b-e; 6e; EV1b, d, e; EV3e, g; EV4a

h) Please note that scale bar and its definition are missing for figure 6c

i) Please give the EV figures a more descriptive title. The information on 'related to' can be given in the legend, e.g.

- Finally, EMBO Reports papers are accompanied online by A) a short (1-2 sentences) summary of the findings and their significance, B) 2-3 bullet points highlighting key results and C) a synopsis image that is 550x300-600 pixels large (width x height) in PNG or JPG format. You can either show a model or key data in the synopsis image. Please note that the size is rather small and that text needs to be readable at the final size. Please send us this information along with the revised manuscript.

- On a different note, I would like to alert you that EMBO Press offers a new format for a video-synopsis of work published with us, which essentially is a short, author-generated film explaining the core findings in hand drawings, and, as we believe, can be very useful to increase visibility of the work. This has proven to offer a nice opportunity for exposure i.p. for the first author(s) of the study.

Please see the following link for representative examples and their integration into the article web page:

<https://www.embopress.org/doi/full/10.15252/embo.2019103932>

With kind regards,

Martina

Referee #1:

The authors investigate how cells adapt/survive ER stress vs succumb/die in response to ER stress. They propose that CHOP is involved in both choices, specifically promoting proliferation to survive ER stress, in addition to its previously established role in promoting cell death during ER stress. As noted before, they developed new mouse CHOP alleles that provide increased temporal resolution to dissect CHOP function, and these new mice may be attractive tools to the research community. A puzzle in the original paper was that CHOP's role in promoting proliferation/survival was seen in their main experimental cell line but not in the NIH3T3 cell line, suggesting that their findings of a new proliferative survival role for CHOP were restricted to selective cell types or perhaps, even unique to their experimental cell line. In the revision, they generate another experimental cell line TIB-73 where CHOP's proliferative effects are also seen. In the revision, they also remove the "negative" data about the NIH3T3 cell line where CHOP's proliferation effects were not important. However, it is important to retain this "negative" data to show that there is clearly cellular variability in CHOP functions, and that the CHOP-dependent proliferation model proposed by the authors is likely to be restricted to specific cells/tissues.

Minor points:

1. In the introduction and discussion, please reference the original papers demonstrating significance of CHOP in diseases as well as those papers where loss of CHOP does not impact diseases. This important variability in CHOP function between cells/tissues is not evident in the 2017 Yang review they cited in lieu of primary literature.
2. In discussion, it would be interesting for authors to speculate how their CHOP-driven proliferation model to survive ER stress applies to neurodegeneration, because neurons are post mitotic and do not divide in response to ER stress.

Referee #2:

I thank the authors for addressing my concerns in their original submission. The findings showcase important roles for CHOP

that depart from its broadly accepted and well-characterized roles in promoting cell death and are an important addition to the field. As such, the new version of the manuscript is suitable for publication in EMBO Reports. I would like to ask the authors to consider including two references to strengthen their discussion further:

1. The authors show that CHOP enhances XBP1 splicing at low ER stress levels (Figs. 6D, E). It has been reported that XBP1 binds the CHOP promoter to induce transcription (PMID 17612490). Thus, it is conceivable that a reinforcing feedback loop is established between CHOP and XBP1S that could contribute to the effects observed.
2. PerturbSeq data has shown precise functional clustering of the genes regulated by each UPR branch (PMID 27984733). This work highlights a feedback loop between IRE1-XBP1 signaling and translocon availability, which could reinforce the notion that a CHOP-XBP1 functional relationship (or more precisely, CHOP-IRE1) is important for restoring ER homeostasis restoration through control of biosynthetic capacity.

Referee #3:

The authors have addressed my primary concerns from the first submission through the inclusion of new experimental data and textual changes. I feel that this is an important paper that will help to 'sway the field' back to view CHOP in a different light that isn't solely focused on its link to apoptosis. I like the idea that CHOP serves as 'stress test' role linking adaptation and death.

Response to Reviewers

In this revised version, we have addressed the remaining concerns of the reviewers as specified below. We have also attended to all of the points raised by the editorial staff in the decision letter—those changes are not detailed here.

Referee #1:

"In the revision, they also remove the "negative" data about the NIH3T3 cell line where CHOP's proliferation effects were not important. However, it is important to retain this "negative" data to show that there is clearly cellular variability in CHOP functions"

The requested "negative" data as well as affirmative data for a role in promoting proliferation in immortalized MEFs (non-3T3s) is now provided in Figure EV1H and EV1I.

"In the introduction and discussion, please reference the original papers demonstrating significance of CHOP in diseases as well as those papers where loss of CHOP does not impact diseases. This important variability in CHOP function between cells/tissues is not evident in the 2017 Yang review they cited in lieu of primary literature."

Citations to the primary literature showing examples of protection by deletion of CHOP and lack thereof have been added to the second-to-last paragraph of the Discussion. We keep the reference to the 2017 Yang review in the Introduction to avoid being unnecessarily duplicative with the Discussion and because that reference provides a fairly broad overview of the evidence in favor of CHOP's apoptotic function in disease.

"In discussion, it would be interesting for authors to speculate how their CHOP-driven proliferation model to survive ER stress applies to neurodegeneration, because neurons are post mitotic and do not divide in response to ER stress."

We have added a brief more explicit mention of how the function of CHOP described in our work might be applicable to non-proliferating cells such as neurons in the second-to-last paragraph of the Discussion.

Referee #2:

"The authors show that CHOP enhances XBP1 splicing at low ER stress levels (Figs. 6D, E). It has been reported that XBP1 binds the CHOP promoter to induce transcription (PMID 17612490). Thus, it is conceivable that a reinforcing feedback loop is established between CHOP and XBP1S that could contribute to the effects observed."

We now include citations to the literature showing regulation of CHOP by ATF6, ATF4, and XBP1 as part of a new larger discussion of how positive feedback and ultrasensitivity within the CHOP axis could contribute to a switch-like function (paragraph beginning "This model proposes that CHOP effectively drives cells into two populations")

"PerturbSeq data has shown precise functional clustering of the genes regulated by each UPR branch (PMID 27984733). This work highlights a feedback loop between IRE1-XBP1 signaling and translocon availability, which could reinforce the notion that a CHOP-XBP1 functional relationship (or more precisely, CHOP-IRE1) is important for restoring ER homeostasis restoration through control of biosynthetic capacity"

Citation to this reference has been added in two locations: in the Results where we describe the effects of CHOP on genes outside of the PERK pathway (last paragraph of the Results) and in the Discussion where we talk about the effects of CHOP on UPR activation beyond just upregulation of chaperones (paragraph beginning "Based on these data, we can put forth a working model to reconcile...")

Dr. D Rutkowski
University of Iowa Carver College of Medicine
United States

Dear Dr. Rutkowski,

I am very pleased to accept your manuscript for publication in the next available issue of EMBO reports. Thank you for your contribution to our journal.

Yours sincerely,
